# Cell-type-specific origins of locomotor rhythmicity at different speeds in larval zebrafish

**Moneeza A Agha[1,2], Sandeep Kishore[1], David L McLean[1,2]\*†**

[1]Department of Neurobiology, Northwestern University, Evanston, United States; [2]Interdisciplinary Biological Sciences Graduate Program, Northwestern University, Evanston, United States

**\*For correspondence:**
dmclean3@ed.ac.uk

**Present address:** †Centre for Discovery Brain Sciences, University of Edinburgh, Edinburgh, United Kingdom

**Competing interest:** The authors declare that no competing interests exist.

**Abstract** Different speeds of locomotion require heterogeneous spinal populations, but a common mode of rhythm generation is presumed to exist. Here, we explore the cellular versus synaptic origins of spinal rhythmicity at different speeds by performing electrophysiological recordings from premotor excitatory interneurons in larval zebrafish. Chx10-labeled V2a neurons are divided into at least two morphological subtypes proposed to play distinct roles in timing and intensity control. Consistent with distinct rhythm generating and output patterning functions within the spinal V2a population, we find that descending subtypes are recruited exclusively at slow or fast speeds and exhibit intrinsic cellular properties suitable for rhythmogenesis at those speeds, while bifurcating subtypes are recruited more reliably at all speeds and lack appropriate rhythmogenic cellular properties. Unexpectedly, however, phasic firing patterns during locomotion in rhythmogenic and non-rhythmogenic V2a neurons alike are best explained by distinct modes of synaptic inhibition linked to cell type and speed. At fast speeds reciprocal inhibition in descending V2a neurons supports phasic firing, while recurrent inhibition in bifurcating V2a neurons helps pattern motor output. In contrast, at slow speeds recurrent inhibition in descending V2a neurons supports phasic firing, while bifurcating V2a neurons rely on reciprocal inhibition alone to pattern output. Our findings suggest cell-type-specific, not common, modes of rhythmogenesis generate and coordinate different speeds of locomotion.

## eLife assessment

In this **fundamental** study, authors present **compelling** evidence for the diversity in cellular and synaptic properties of one class of spinal interneurons and tie it to their differentiated role in locomotor pattern generation. The findings reported here will be of broad interest to neuroscientists in general and to motor systems scientists in particular.

## Introduction

All forms of vertebrate locomotion rely on rhythmic activity in muscles operating as functional antagonists (*Grillner, 1985*). For instance, swimming in fishes relies on left–right alternation of axial muscles across the body, while walking in mammals relies on flexor–extensor alternation along it. A landmark hypothesis posed over a century ago argued for corresponding excitatory 'half-centers' within spinal cord (e.g., left–right or flexor–extensor), with reciprocal, feedforward inhibition between half-centers generating rhythmic output (*Brown, 1911*). Since then reciprocal inhibition has been relegated to patterning activity between so-called 'unit burst generators' capable of rhythmogenesis via recurrent, feedback excitatory connections (*Grillner and Kozlov, 2021*).

The idea of solitary unit burst generators reliant on recurrent connectivity disconnects rhythmogenesis from activity in other unit burst generators, which helps explain the flexible patterns of coordination necessary to effectively navigate through unpredictable environments. For example, simulations adjusting the synaptic coupling strength between segmental axial unit burst generators can explain changes in the speed and direction of swimming in lampreys (*Kozlov et al., 2009*). Likewise, simulations altering the synaptic coupling between segmental limb unit burst generators can replicate different speeds and gaits of locomotion in cats (*Grillner, 1981*).

Another theory explaining speed-dependent changes in coordination stratifies 'rhythm' and 'pattern' control (*Perret and Cabelguen, 1980*; *Koshland and Smith, 1989*; *Kriellaars et al., 1994*; *Burke et al., 2001*; *Lafreniere-Roula and McCrea, 2005*), where populations of interneurons generating higher- or lower-frequency rhythms feed into separate populations coordinating distinct patterns of motor output (*McCrea and Rybak, 2008*; *Ausborn et al., 2019*). In support, studies in zebrafish and mice suggest separate spinal populations are active at different locomotor speeds with distinct patterns of motor output (*McLean et al., 2008*; *Talpalar et al., 2013*; *Satou et al., 2020*; *Picton et al., 2022*). Critically, however, rhythmogenesis is presumed to originate from a common mechanism regardless of speed, reliant on cellular properties or synaptic drive or both (*Guertin, 2009*; *Brocard et al., 2010*; *Ryczko et al., 2010*; *Ziskind-Conhaim and Hochman, 2017*; *El Manira, 2023*).

Here, we have explored the origins of spinal rhythmogenesis at different speeds using the larval zebrafish model system, where it is possible to link the identity, physiology and connectivity of interneurons to their recruitment patterns at different speeds of locomotion. Like all fish, larval zebrafish generate two basic modes of forward swimming or 'gaits' (*Di Santo et al., 2021*). Anguilliform mode is typically reserved for escaping predators and uses higher-frequency/-amplitude oscillations of the trunk, while carangiform mode utilizes lower-frequency/-amplitude oscillations of the tail for exploration and pursuing prey (*Budick and O'Malley, 2000*; *Müller and van Leeuwen, 2004*; *Thorsen et al., 2004*; *Patterson et al., 2013*; *Bhattacharyya et al., 2021*).

Swimming at different speeds is generated by axial motor neurons organized into fast, slow, and intermediate units distinguished by birth order (*Myers et al., 1986*), size (*McLean et al., 2007*), relative innervation of fast and slow muscle fibers (*Bello-Rojas et al., 2019*), and molecular signatures (*D'Elia et al., 2023*; *Pallucchi et al., 2024*). Current- and voltage-clamp recordings from motor neurons have revealed differences in rhythmogenic cellular properties and synaptic drive between the different types (*Menelaou and McLean, 2012*; *Kishore et al., 2014*), suggesting different forms of rhythmogenesis could be responsible for different speeds of locomotion. To test this idea, we performed current- and voltage-clamp recordings from Chx10-labeled spinal V2a interneurons, which provide a major source of phasic premotor excitation during locomotion in zebrafish and mice (*McLean and Dougherty, 2015*).

We find that morphologically distinct V2a neurons sort into rhythm-generating and pattern-forming subtypes based on distinct cellular properties linked to distinct patterns of recruitment and synaptic drive. Critically, among descending V2a neurons that qualify as rhythm-generating, reciprocal inhibition originating from contralateral sources supports phasic firing patterns at fast speeds, while recurrent inhibition from ipsilateral sources supports phasic firing patterns at slow speeds. For bifurcating V2a neurons considered pattern-forming, the opposite is true – recurrent inhibition supports phasic firing at fast speeds and reciprocal inhibition at slow speeds. Ultimately, our findings reveal unexpected cell-type-specific and speed-dependent complexity in the mechanisms of spinal rhythmogenesis underlying locomotion.

## Results
### Identifying V2a neurons for recordings during 'fictive' escape swimming

From our previous work studying V2a neurons in larval zebrafish we know they are organized into: (1) fast and slow subtypes based on dorsoventral (DV) soma position and birth order (*McLean et al., 2007*; *McLean et al., 2008*; *McLean and Fetcho, 2009*), and (2) distinct subtypes that provide stronger direct versus indirect control of motor neuron recruitment (*Menelaou et al., 2014*; *Menelaou and McLean, 2019*). Specifically, V2a neurons with descending axons (V2a-D) provide stronger inputs to other premotor interneurons, while V2a neurons with bifurcating axons (V2a-B) provide stronger

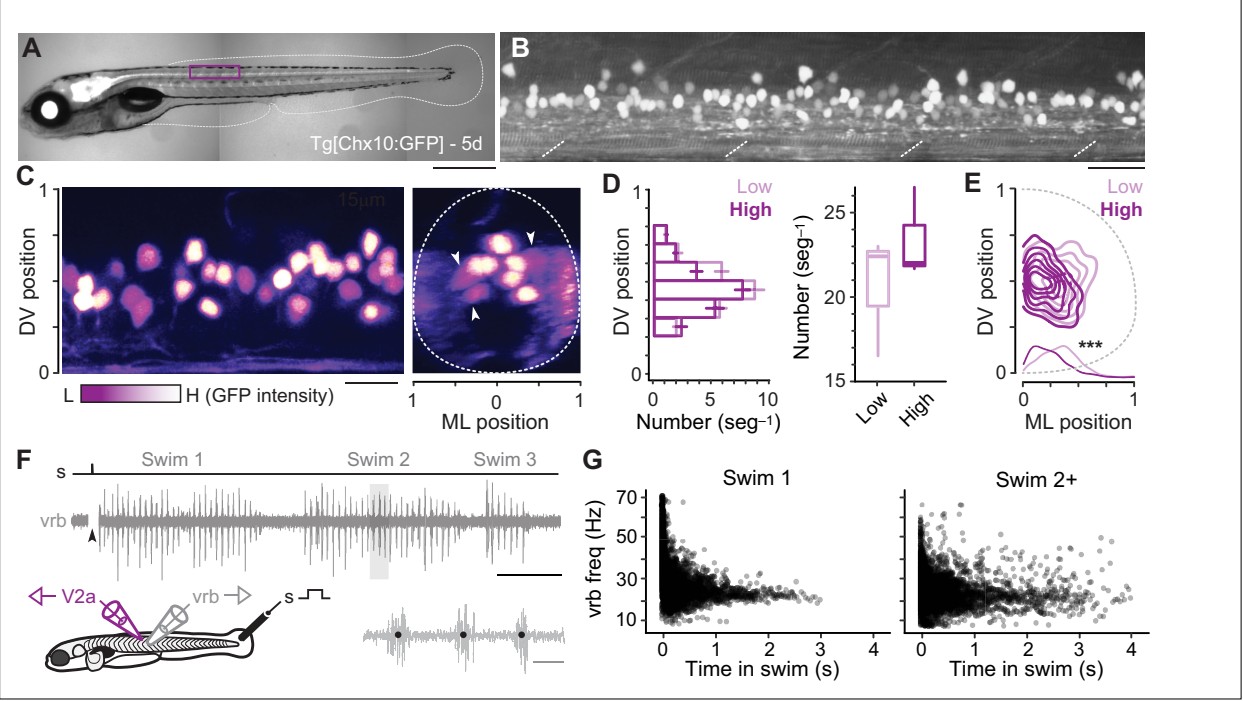

**Figure 1.** Identification of V2a neurons for recordings during 'fictive' swimming. (**A**) Composite differential interference contrast (DIC)/epifluorescence image of Tg[Chx10:GFP] larval zebrafish at 5 days post fertilization (5d). Dorsal is up and rostral is left. Scale bar, 0.5 mm. (**B**) Confocal image of the Tg[Chx10:GFP] spinal cord from segments 10–15 at midbody (purple box in *A*). Scale bar, 30 µm. (**C**) Pseudo-colored images of V2a somata based on high (H, white) and low (L, purple) GFP intensity. Left panel: sagittal section, right panel: coronal section. Dashed lines indicate boundaries of spinal cord, which are normalized to 0–1 in the dorsoventral (DV) and mediolateral (ML) axes. White arrowheads indicate large, lateral V2a somata with low levels of GFP expression. Scale bar, 20 µm. (**D**) Plots of numbers of low- and high-intensity V2a neurons per midbody segment (*n* = 5–6 segments from 3 fish). Left panel: bar plot of average number (± standard error of the mean or SEM) of low and high GFP V2a neurons along the DV axis. There is no significant difference in the medians of the DV distributions of high and low GFP expressing V2a neurons (Wilcoxon rank-sum rest; *W* = 684, p = 0.698, *n* = 347 high and 401 low GFP neurons from 3 fish). Right panel: box and whisker plot showing average number of high and low GFP neurons per segment (Wilcoxon rank-sum test; *W* = 88, p = 0.053, *n* = 17 segments in 3 fish). (**E**) Two-dimensional contour density plots showing the ML and DV distributions of V2a neurons expressing low or high levels of GFP. Density distributions of ML positions of V2a neurons are shown at the bottom of the contour plots, which are significantly different (***, two-sample Kolmogorov–Smirnov test; *D* = 0.164, *p* < 0.001, *n* = 347 high and 401 low GFP neurons from 3 fish). (**F**) Top panel: an example of an extracellular ventral rootlet recording of three consecutive swim episodes evoked by a mild electrical stimulus (s) to the tail fin (at arrowhead, artifact blanked). Scale bar, 500 ms. Bottom left panel: a cartoon of the locations of the stimulus and the recording electrodes. Bottom right panel: individual ventral rootlet bursts (vrb) on an expanded timescale. Black dots mark the center of each vrb used to calculate swim frequency in Hz (1/s), which is indicative of swimming speed. Scale bar, 20 ms. (**G**) Scatter plot of vrb frequency (Hz) versus the time in swim episode (ms). The swimming episode immediately following the stimulus is called Swim 1 (left), and any subsequent swimming episodes are called Swim 2+ (right).

inputs to motor neurons (*Menelaou and McLean, 2019*), in line with differential rhythm and pattern control. However, we do not know if the same mechanisms of rhythmogenesis shape the phasic activation of distinct V2a types at different speeds.

To record from distinct types of V2a neurons, we targeted green fluorescent protein (GFP)-positive neurons located at midbody in the Tg(Chx10:GFP) line (*Figure 1A, B*; *Kimura et al., 2006*), since midbody neurons would be recruited during both fast anguilliform and slow carangiform modes (*Müller and van Leeuwen, 2004*; *Thorsen et al., 2004*). V2a-D and V2a-B neurons recruited at fast speeds can be distinguished by levels of Chx10 expression in transgenic lines (*Menelaou and McLean, 2019*), as reported in mice (*Hayashi et al., 2018*). To get a better sense of differences in the number and spatial distribution of distinct types, we split GFP-positive neurons into two categories based on a bimodal distribution of fluorescence intensity and normalized their soma positions to DV and mediolateral (ML) axes in the spinal cord. Larger, dimmer V2a neurons intermingled with smaller, brighter neurons along the DV axis (*Figure 1C*, *left*), but occupied more lateral positions (*Figure 1C*, *right*). Statistical analysis confirmed these observations, with no significant difference in either DV distribution (*Figure 1D*, *left*) or number per segment between the low or high GFP-positive types (*Figure 1D*, *right*), but a significant difference in ML position (*Figure 1E*).

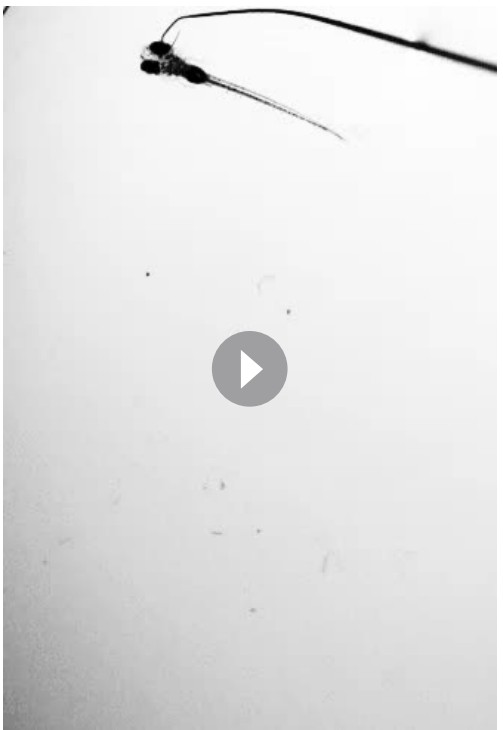

**Video 1.** High-speed video of real escape swimming in a 5-day-old larva. (A) Larva is touched near the head by a sharpened tungsten pin, which triggers a rapid escape bend, followed by high-frequency anguilliform swimming and then ending with low-frequency carangiform swimming. Video was captured at 2000 frames per second and has been slowed down 40× for purposes of visualization (playback at 50 frames per second).

https://elifesciences.org/articles/94349/figures#video1

To evaluate firing patterns and synaptic drive responsible for rhythmogenesis in different types, we recorded from V2a neurons during 'fictive' swimming evoked by a mild electrical shock, while simultaneously monitoring rhythmic motor output using ventral rootlet recordings (*Figure 1F*). During real swimming escape responses decelerate (*Video 1*), beginning in fast anguilliform mode and ending in slow carangiform mode (*McLean et al., 2008*; *McLean and Fetcho, 2009*). Plots of ventral rootlet burst frequencies with time confirm this pattern during 'fictive' escape responses, where we observe the highest frequencies at the beginning of the swim episode immediately following the stimulus (*Figure 1G*, *left*). Notably, zebrafish swimming is episodic (*McHenry and Lauder, 2006*; *Soto and McHenry, 2020*) and so it can continue intermittently after the initial escape swim episode. Subsequent episodes (Swim 2+) rarely exceeded peak firing frequencies observed in the first stimulus-evoked episode (>60 Hz in Swim 1) and there was more variability in frequency (*Figure 1G*, *right*). For purposes of analysis, we pooled data from the first (Swim 1) and subsequent (Swim 2+) episodes to capture as broad a frequency range as possible.

Thus, V2a neurons can be easily distinguished for targeting purposes using levels of GFP expression and we can assess their firing and synaptic drive during fast anguilliform and slow carangiform modes distinguished by frequency and temporal sequence within fictive escape swim episodes. This set the stage for asking if the same mechanism of rhythmogenesis drives the phasic activation of distinct V2a types at different speeds.

## Differences in intrinsic rhythmogenesis related to V2a neuron type

Rhythmogenesis can originate from cellular properties or synaptic drive or some combination of the two (*Marder and Calabrese, 1996*; *Marder and Bucher, 2001*). In theory, neurons that are intrinsically rhythmogenic can create patterned outputs from unpatterned inputs. Thus, to explore the intrinsic rhythmicity of V2a neurons we performed whole-cell current-clamp recordings from high and low GFP-positive V2a neurons and tested their firing responses to stable current steps. By including dye in our recording pipette, we confirmed post hoc that all high GFP neurons we targeted for recordings were V2a-Ds and all low GFP neurons were V2a-Bs (*Figure 2A*), as we would expect based on our prior work (*Menelaou and McLean, 2019*). Differences in sizes and DV distributions of V2a-D and V2a-B neurons in our current-clamp dataset (*n* = 47) were consistent with our analysis of high and low GFP neurons in stable lines (*Figure 2B*, *left*). Also, measurements of input resistance and rheobase current were consistent with differences in sizes and excitability between the two types (*Figure 2C*, *left*). Thus, we were confident of a sufficient sampling of V2a neurons to explore differences in rhythmogenic capacity related to type and recruitment order.

V2a-D neurons with 'tonic' responses and V2a-B neurons with higher-frequency 'chattering' ones were distributed dorsally and had the lowest input resistances in their respective types (*Figure 2B–E*). In tonic V2a-Ds, firing above rheobase current was observed consistently throughout the current step (*Figure 2E*, *top*), with instantaneous spike rates increasing linearly with increasing current, starting around 40 Hz and reaching 200 Hz (*Figure 2D*). In chattering V2a-Bs, firing above rheobase current

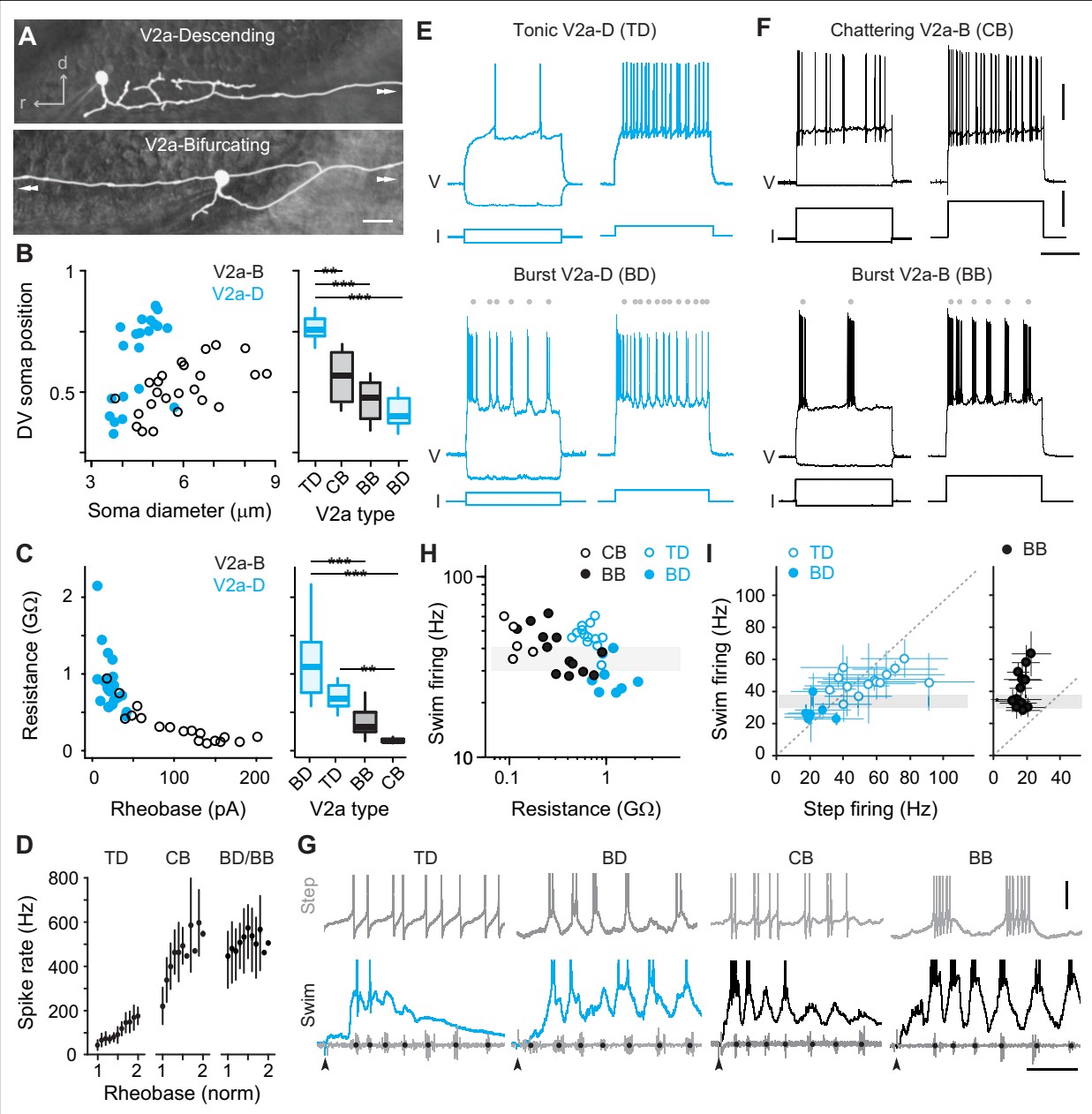

**Figure 2.** Cell-type-specific electrophysiological properties linked to recruitment order. (**A**) Composite DIC/epifluorescent images of post hoc fills of V2a-Descending (V2a-D, top) and V2a-Bifurcating (V2a-B, bottom) neurons. Arrows indicate continuation of axons outside the field of view. Scale bar, 10 μm. (**B**) Left panel: scatter plot of soma diameter of V2a-D and V2a-B neurons normalized to dorsoventral (DV) soma position, with 1 demarcating the dorsal and 0 the ventral boundaries of the spinal cord. The soma diameters of V2a-Ds (n = 22) and V2a-Bs (n = 25) are significantly different according to the Wilcoxon rank-sum test (W = 401, p < 0.001). Right panel: box plots of DV soma positions of V2a-D and V2a-B neurons classified as tonic, bursting, or chattering based on their step-firing responses in panels *E* and *F*. TD = tonic V2a-D (n = 14), CB = chattering V2a-B (n = 12), BB = bursting V2a-B (n = 13), BD = bursting V2a-D (n = 8). The differences in the DV soma positions of TD and CB (z = −3.202, p < 0.01), TD and BB (z = −4.714, p < 0.001), and TD and BD (z = −4.892, p < 0.001) are significantly different according to Dunn's test following post hoc Benjamini–Hochberg correction. (**C**) Left panel: scatter plot of input resistance (GΩ) versus the rheobase (pA) of V2a-B (black) and V2a-D (blue) neurons. The input resistance (W = 30, p < 0.001) and rheobase (W = 361.5, p < 0.001) of V2a-Bs (n = 18) and V2a-Ds (n = 21) are significantly different according to the Wilcoxon rank-sum test. Right panel: box plots of input resistance (GΩ) of BD (n = 7), TD (n = 14), BB (n = 11), and CB (n = 7) V2a neurons. The differences in the input resistance of BD and BB (z = −3.531, p < 0.001), BD and CB (z = 4.734, p < 0.001), and CB and TD neurons (z = −3.559, p < 0.01) are significantly different according to Dunn's test following post hoc Benjamini–Hochberg correction. (**D**) Mean (± standard deviation [SD]) instantaneous spike rates (Hz) between 1 and 2x rheobase for TD, CB, BD, and BB neurons. (**E**) Top panel: step-firing responses of V2a-D neurons defined as tonic (TD). Bottom panel: step-firing responses of V2a-D neurons defined as bursting (BD). Gray dots depict the midpoint of slower membrane oscillations driving the bursting spiking behavior. Same

*Figure 2 continued*

scale as panel *F*. (**F**) Top panel: step-firing responses of V2a-B neurons defined as chattering (CB). Bottom panel: V2a-B neurons defined as bursting (BB). Gray dots depict the midpoint of slower membrane oscillations driving the bursting spiking behavior. Scale bars, 20 mV, 200 pA, 200 ms. (**G**) Top panels: step-firing responses close to rheobase of TD, BD, CB, and BB neurons. Bottom panels: swim-firing responses evoked by a mild electrical shock shown on the same time scales. Black dots on ventral root bursts (gray) indicate swim frequency. Black arrows mark the stimulation artifact. Scale bars, 10 mV, 50 ms. (**H**) Scatter plot of median swim-firing frequency (Hz) versus the input resistance (GΩ) of CB, TD, BB, and BD on logarithmic *x*- and *y*-scales. Shaded gray box indicates 30–40 Hz range reflecting transition between anguilliform and carangiform swim modes. There is a significant negative correlation between the median swim-firing frequency and input resistance of TD and BD V2a neurons (Spearman's rank correlation test, $r_s$ = −0.723, p < 0.001, *n* = 21) as well as CB and BB V2a neurons ($r_s$ = −0.624, p < 0.01, *n* = 18). (**I**) A comparison of median swim-firing and median step-firing frequencies between 1 and 2x rheobase (± SD) for TD, BD, and BB neurons. For TD neurons, step frequencies represent spike rates, while for BD and BB neurons they represent burst rates. CB neurons were too variable in step-firing frequency to obtain reliable median values. Shaded gray box indicates 30–40 Hz range reflecting transition between anguilliform and carangiform swim modes. There is a significant positive correlation between the median swim-firing and median step-firing frequencies of TD and BD neurons (Spearman's rank correlation test; $r_s$ = 0.768, p < 0.001, *n* = 21), but not BB neurons (Spearman's rank correlation test; $r_s$ = 0.318, p = 0.289, *n* = 13). Statistically significant differences are denoted as follows: *p < 0.05; **p < 0.01; ***p < 0.001.

was intermittent throughout the step (*Figure 2F*, *top*), with instantaneous rates starting around 200 Hz and reaching 600 Hz (*Figure 2D*). These observations are consistent with previous recordings from fast V2a-D and V2a-B neurons (*Menelaou and McLean, 2019*).

On the other hand, V2a-D and V2a-B neurons with 'burst' responses were distributed ventrally and had the highest input resistances within their respective types (*Figure 2B–E*). In bursting V2a neurons, firing above rheobase was characterized by higher-frequency spikes superimposed on lower-frequency membrane potential oscillations (*Figure 2E, F*, *bottom*). While the frequency of the slower oscillations increased with current (*Figure 2E, F*, *bottom*), instantaneous spike rates were already close to peak values at rheobase (500–600 Hz; *Figure 2D*). Beyond 2–3x rheobase current, slower oscillations were no longer obvious and the firing pattern more closely resembled the chattering one (data not shown), as observed previously for motor neurons (*Menelaou and McLean, 2012*).

Next, we compared the firing properties of distinct types of V2a neurons to their recruitment patterns during fictive escape swimming. As expected (*Menelaou and McLean, 2019*), both tonic V2a-D and chattering V2a-B neurons were recruited at the beginning of the episode at median swim frequencies above 30 Hz (*Figure 2G, H*), consistent with recruitment during fast anguilliform mode (*Thorsen et al., 2004*; *McLean et al., 2008*; *McLean and Fetcho, 2009*). In contrast, burst V2a-Ds fired more reliably at the end of the episode at median frequencies that rarely exceeded 30 Hz (*Figure 2G, H*), consistent with participation in slow carangiform mode (*Thorsen et al., 2004*; *McLean et al., 2008*; *McLean and Fetcho, 2009*). Burst V2a-Bs were recruited just below 30 Hz, and fired more reliably throughout the episode and over a broader frequency range (*Figure 2G, H*), particularly compared to the narrow range of bursting V2a-Ds (two sample Kolmogrov–Smirnov test on density distributions; D = 0.157, p < 0.001; BB, *n* = 6, BD, *n* = 8). A comparison between the step- and swim-firing patterns among V2a-D neurons revealed a remarkable correspondence (*Figure 2G*), with median spike frequencies (tonic) or oscillation frequencies (burst) overlapping median recruitment frequencies during swimming (*Figure 2I*, *left*). However, the link between step- and swim-firing patterns was less obvious for V2a-B neurons (*Figure 2G*). For example, while bursting V2a-B neurons recruited at swim frequencies exceeding 60 Hz had higher-frequency oscillations, step bursting rarely exceeded 20 Hz (*Figure 2I*, *right*). For chattering V2a-B neurons, there were no clear subthreshold oscillations to calculate rhythmicity and the broad range of spike frequencies resulted in large standard deviations (±74–130 Hz, *n* = 5) that rendered median values less interpretable.

Thus, distinct electrophysiological properties define distinct V2a types recruited at different speeds and V2a-D neurons are better suited to providing intrinsic rhythmogenic activity during fast and slow swimming modes. This is consistent with their proposed rhythm-generating and pattern-forming roles (*Menelaou and McLean, 2019*), and suggests V2a-B neurons rely more on rhythmic synaptic drive to shape their phasic firing patterns during swimming.

## Both types of V2a neuron operate at slow and fast speeds

Our current-clamp data suggest that different types would rely differently on cellular properties versus synaptic drive for rhythmogenesis. To test this idea more directly, we performed voltage-clamp recordings during fictive escape swimming. To more effectively isolate synaptic currents, we included cesium in the patch solution. Since this blocks voltage-gated channels and spiking, we needed to classify

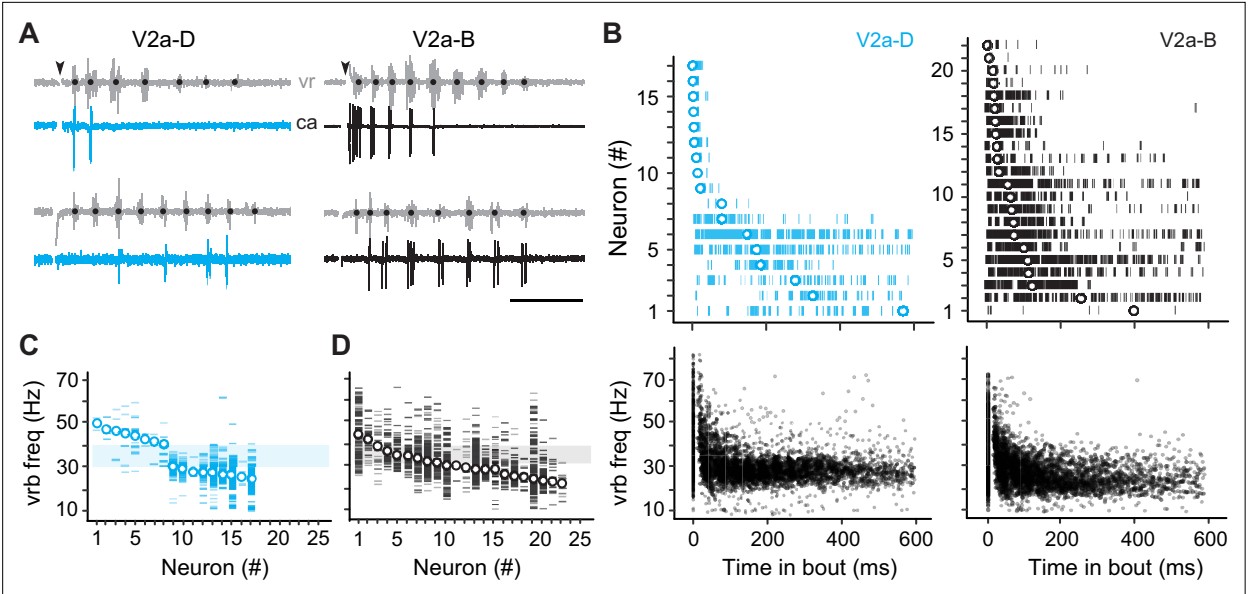

**Figure 3.** Differences in recruitment patterns among the distinct V2a types. (**A**) Left panel: cell-attached (ca) recordings of descending V2a (V2a-D) neurons during fictive swimming evoked by a brief stimulus (at black arrowhead). Black dots on ventral root (vr) bursts (gray) indicate swim frequency. The top neuron fires immediately after the stimulus during high-frequency swimming, while the bottom neuron fires near the end of the episode at lower frequencies. Right panel: as shown to the left, but for bifurcating V2a (V2a-B) neurons. Scale bar, 100 ms. (**B**) Top panel: spike timing of V2a-D (left) and V2a-B (right) neurons relative to the start of swimming ordered by the median of the distribution. Bottom panel: a scatterplot of vrb frequency (Hz) as a function of time for all episodes (Swim 1 and 2+) during which the V2a-D (left) or V2a-B (right) neurons were recruited. (**C**) Raster plots of ventral root burst frequencies (Hz) over which V2a-Ds were recruited ordered by the median vrb recruitment frequency (Hz). Shaded box indicates 30–40 Hz range reflecting transition between anguilliform and carangiform swim modes. (**D**) Raster plots of ventral root burst frequencies (Hz) over which V2a-Bs were recruited ordered by the median vrb recruitment frequency (Hz). Shaded box indicates 30–40 Hz range reflecting transition between anguilliform and carangiform swim modes.

fast and slow V2a-D and V2a-B neurons subtypes by first monitoring spikes in cell-attached mode. Consistent with our current-clamp recordings, in our voltage-clamp dataset (*n* = 39) we found V2a-D and V2a-B neurons that fired at the beginning of the escape episode or nearer the end (*Figure 3A*). V2a-D neurons fired more sparsely (*Figure 3A*, *left*) and were divided into two distinct groups: (1) neurons that fired exclusively at the beginning of the episode at median swim frequencies exceeding 40 Hz, and (2) neurons that fired later on in the episode at median swim frequencies rarely exceeding 30 Hz (*Figure 3B*, *top and bottom left*; *Figure 3C*). On the other hand, V2a-B neurons fired more reliably (*Figure 3A*, *right*) and median values were more continuous, making it more difficult to group neurons based on location within the episode (*Figure 3B*, *top and bottom right*) or swim frequency (*Figure 3D*).

To better segregate both types of V2a neurons based on speed, we used the frequency-dependent transition between anguilliform and carangiform modes to define V2a neurons recruited more reliably below 35 Hz as 'slow' and V2a neurons recruited more reliably above 35 Hz as 'fast' (*Figure 4A*). Consistent with our current-clamp recordings, slow V2a-D and V2a-B neurons in our voltage-clamp recordings were more ventrally distributed with higher resistances than fast V2a-D and V2a-B neurons (*Figure 4B*). A closer examination of spike timing relative to individual ventral rootlet bursts revealed median spike distributions in advance of the center of the motor burst, as expected for premotor excitatory interneurons involved in rhythmogenesis (*Figure 4C*). Plots of spike densities normalized to center burst revealed that V2a-B neurons begin to fire slightly in advance of V2a-D neurons (*Figure 4D*), consistent with differences in membrane resistance related to size, where neurons with faster time constants respond more quickly to synaptic inputs (*Menelaou and McLean, 2019*).

Next, to explore recruitment probability as a function of speed in more depth we divided the full range of fictive swimming frequencies into four bins distributed around 35 Hz. For V2a-D neurons, fast neurons were rarely recruited below 35 Hz and increased to a maximum of 50% recruitment reliability per motor burst above 35 Hz (*Figure 4E*, *top left*). Similarly, slow V2a-D neurons did not exceed 50%

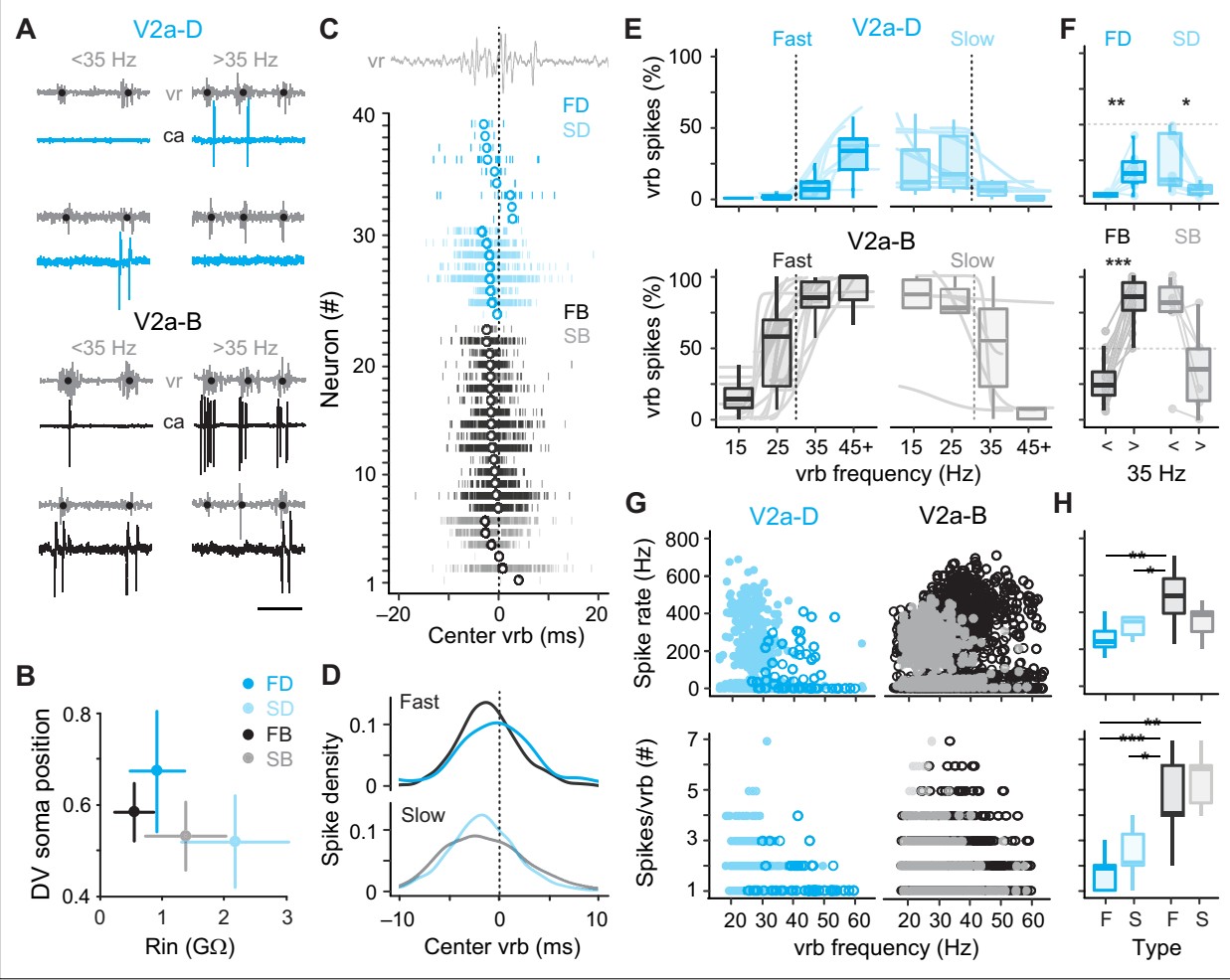

**Figure 4.** Distinct types of slow and fast V2a neurons distinguished based on recruitment. (**A**) Top panel: cell-attached (ca) and ventral rootlet (vr) recordings during slow (<35 Hz) and fast (>35 Hz) swimming illustrate V2a-D neurons (blue) that fire exclusively at slow and fast speeds. Bottom panel: V2a-B neurons (black) fire more reliably at both slow and fast speeds. Scale bar, 25 ms. (**B**) Plots of the mean (± standard deviation [SD]) dorsoventral (DV) positions of fast (FD) and slow (SD) V2a-Ds, and fast (FB) and slow (SB) V2a-Bs versus their mean (± SD) input resistance (Rin). (**C**) Spike timing of slow (S) and fast (F) V2a-D and V2a-B neurons relative to the center of the ventral root burst (vrb) depicted by the dashed vertical line. Tick marks represent individual spikes from multiple cycles and circles represent median spike timing. (**D**) Density plots of spike timing relative to the center burst for slow and fast V2a-B and V2a-D neurons illustrated in panel *C*. (**E**) Top panel: box plots of the percentage of ventral root bursts with spikes (vrb spikes %) as a function of vrb frequency (Hz) for fast (FD, *n* = 9) and slow (SD, *n* = 8) V2a-D neurons. For box plots, the ventral root burst frequency was binned at 15 Hz intervals starting at 15 Hz, and the final interval of 45+ Hz consisted of a 20-Hz range spanning from 45 to 65 Hz. Box plots are superimposed on trendlines from individual neurons fit to data binned at 5 Hz intervals, whose slopes define whether they are fast (positive) or slow (negative). Dashed line indicates transition between slow carangiform and fast anguilliform modes. Bottom panel: as above but for fast (FB, *n* = 16) and slow (SB, *n* = 6) V2a-B neurons. (**F**) Top panel: box plots of vrb spikes % for slow and fast V2a-D neurons from panel E collapsed into single slow (<35 Hz) and (>35 Hz) bins for purposes of statistical analysis (Wilcoxon signed-rank test; FD, *V* = 1, p < 0.01, *n* = 9; SD, *V* = 36, p = <0.01, *n* = 8). Bottom panel: as above but for V2a-B neurons (Wilcoxon signed-rank test; FB, *V* = 0, p < 0.001, *n* = 16; SB, *V* = 20, p = 0.063, *n* = 6). (**G**) Top panel: scatter plots of instantaneous spike rates versus vrb frequency (Hz) for V2a-D (left) and V2a-B (right) neurons color coded as fast (dark) and slow (light). FD, *n* = 9; SD, *n* = 8; FB, *n* = 16; SB, *n* = 6. Bottom panel: scatter plots of the number of spikes per ventral root burst (#) as a function of vrb frequency (Hz), organized and color coded as above. (**H**) Top panel: box plots of maximum instantaneous spike rates for fast (*n* = 5) and slow (*n* = 7) V2a-D and fast (*n* = 16) and slow (*n* = 5) V2a-B neurons. Note, only a subset of V2a neurons that fired two or more spikes per cycle were included in this analysis (Wilcoxon rank-sum test; FB-FD, *W* = 73, p < 0.01; FB-SD, *W* = 90, p < 0.05). Bottom panel: box plots of the maximum number of spikes/vrb (Wilcoxon rank-sum test; FB-FD, *W* = 133, p < 0.001; FB-SD, *W* = 100, p < 0.05; FD-SB, *W* = 50, p < 0.01; FD, *n* = 9; SD, *n* = 8; FB, *n* = 16; SB, *n* = 6). Statistically significant differences are denoted as follows: *p < 0.05; **p < 0.01; ***p < 0.001.

recruitment reliability per motor burst below 35 Hz and rarely fired above it (*Figure 4E*, *top right*). When collapsed into two bins below and above 35 Hz, these differences in recruitment probability were significant (*Figure 4F*, *top*). For V2a-B neurons, fast neurons fired more reliably below 35 Hz than fast V2a-Ds and increased to 100% recruitment reliability per motor burst above 35 Hz (*Figure 4E*, *bottom left*). Similarly, slow V2a-B neurons fired more reliably above 35 Hz than slow V2a-Ds, and at 100% reliability below it (*Figure 4E*, *bottom right*). Binning of recruitment among V2a-Bs revealed significant differences related to speed among the fast subtypes only (*Figure 4F*, *bottom*). Among the slow V2a-Bs, decreases in recruitment reliability were observed in the majority of neurons (*n* = 5 of 6), however the difference was not significant when pooled (*Figure 4F*, *bottom*).

In further support of the speed-based modular organization of V2a neurons, in slow types higher instantaneous spike rates were observed at lower burst frequencies, while in fast types higher spike rates were observed at higher burst frequencies (*Figure 4G*, *top*). Critically, spike rates observed during fictive swimming overlap with those observed during current steps between 1 and 2x rheobase, ranging from in 200–600 Hz (*Figure 4H*, *top*), with the fastest spike rates observed in fast V2a-B neurons. This suggests that the firing patterns observed during current injections between 1 and 2x rheobase are functionally relevant. When we measured spike numbers per burst, lower numbers were consistently observed at higher burst frequencies regardless of type or speed class (*Figure 4G*, *bottom*) consistent with less time to spike during shorter cycle periods. In addition, V2a-B neurons fired significantly more spikes per burst than V2a-Ds on average (*Figure 4H*, *bottom*), consistent with their higher recruitment probabilities.

Thus, within our voltage-clamp dataset we have V2a-D and V2a-B neurons operating during anguilliform and carangiform swimming modes. While V2a-D neurons fire more sparsely and in a stricter modular fashion, V2a-B neurons fire more reliably and over a broader range of frequencies, consistent with occupation of a pattern-forming layer downstream of distinct fast and slow rhythmic signals.

## Differences in the levels of synaptic drive mirror differences in speed modularity

To get a better sense of the relative segregation of synaptic inputs to V2a-D and V2a-B neurons based on speed, we first focused on the maximum amplitude of excitatory and inhibitory currents recorded during fictive escape swimming. Clamping V2a neurons at a potential to isolate excitatory postsynaptic currents (EPSCs) revealed faster time course, compound EPSCs superimposed on a slower, tonic inward current (*Figure 5A, B*). While tonic excitation was observed the entirety of the episode, fast V2a-Ds received compound EPSCs exclusively during high-frequency swimming at the beginning (*Figure 5A*, *top*), while slow V2a-D's received compound EPSCs exclusively during slower swimming later in the episode (*Figure 5A*, *bottom*), consistent with their recruitment patterns. A similar pattern was observed among V2a-Bs, however compound EPSCs were observed over a broader range of frequencies (*Figure 5B*), also consistent with their recruitment patterns. We also noted differences in maximum levels of current related to type. When we plotted maximum current (tonic current + compound EPSCs) versus input resistance, we observed a significant negative correlation, with lower resistance V2a-Bs receiving orders of magnitude more excitation than higher resistance V2a-Ds (*Figure 5C*). This is consistent with peak excitatory drive normalized to cellular excitability, as reported previously (*Menelaou et al., 2022*).

Next, clamping V2a neurons at a potential to isolate inhibitory post-synaptic currents (IPSCs) revealed faster time course compound IPSCs superimposed on a slower, tonic outward current (*Figure 5D, E*). In all V2a types tonic inhibition persisted the entirety of the episode and compound IPSCs were larger at higher frequencies near the beginning (*Figure 5D, E*). Like excitation, plots of maximum current (tonic current + compound IPSCs) revealed a significant negative correlation between inhibitory current and input resistance (*Figure 5F*), meaning that peak inhibitory drive is also normalized to cellular excitability. Given these large differences in maximum current values, to compare frequency-dependent changes in excitation and inhibition, we normalized the currents for each cycle to the maximum currents for the neuron (*Figure 5C, F*). So, maximum EPSCs or IPSCs in a given cycle are expressed as a percentage of the maximum value in all cycles at any speed.

For fast V2a-D and V2a-B neurons, maximum excitatory drive increased with increasing frequency (*Figure 5G*, *left*). Differences in maximum excitation were significant for both types when data were pooled (*Figure 5H*), consistent with their recruitment patterns (*Figure 4F*). Inhibition also followed the

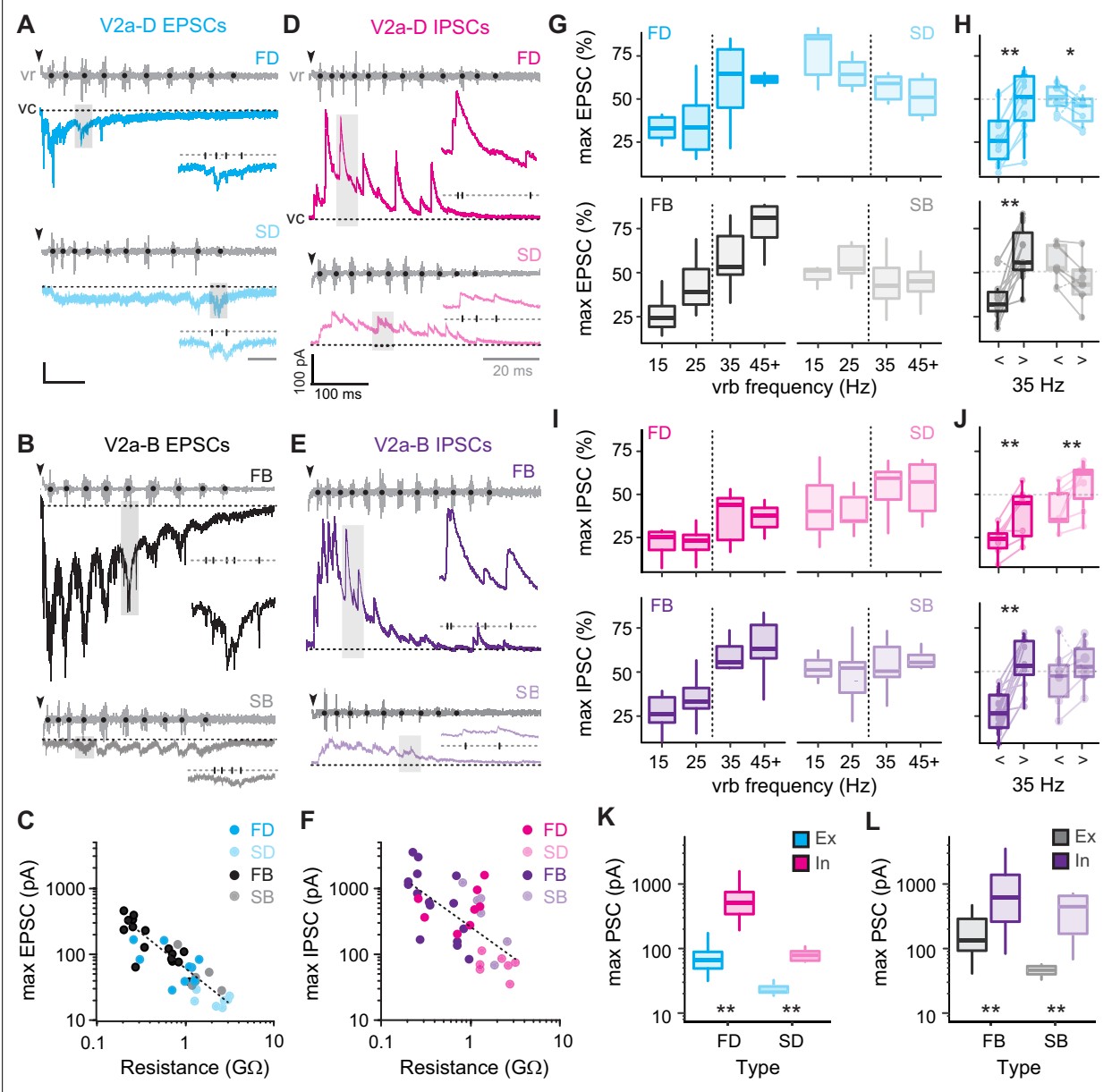

**Figure 5.** Differences in levels of synaptic drive related to speed and V2a cell type. (**A**) Voltage-clamp (vc) recordings of excitatory post-synaptic currents (EPSCs) received by fast (top, FD) and slow (bottom, SD) V2a-D neurons along with ventral root (vr) recordings during fictive swimming triggered by an electrical stimulus (black arrowheads, artifact blanked). Black dots indicate burst intervals for frequency measures. Gray shaded boxes indicate region expanded inset, illustrating the holding potential (dashed line, −75 mV) and individual EPSCs (tick marks). Scale bars, 25 pA, 100 ms (20 ms inset). (**B**) As in *A*, but for fast (top, FB) and slow (bottom, SB) V2a-B neurons. (**C**) Scatter plot of maximum EPSC amplitude (pA) as a function of input resistance (GΩ) on logarithmic *x*- and *y*-scales. Dashed logarithmic trendlines are included for illustrative purposes (Spearman's rank correlation test, $r_s$ = −0.873, p < 0.001, n = 39). (**D**) Voltage-clamp recordings of inhibitory post-synaptic currents (IPSCs) at a holding potential of 10 mV from fast (top, FD) and slow (bottom, SD) V2a-D neurons, organized as detailed in panel A. Scale bars, 100 pA, 100 ms (20 ms inset). (**E**) As in D, but fast (top, FB) and slow (bottom, SB) V2a-B neurons. (**F**) Scatter plot of maximum IPSC amplitude as a function of input resistance (GΩ) on logarithmic *x*- and *y*-scales (Spearman's rank correlation test; $r_s$ = −0.633, p < 0.001, n = 36). (**G**) Top panel: box plots of the maximum EPSC per cycle as a percentage of the maximum current (max EPSC%) for fast (FD, *n* = 9) and slow (SD, *n* = 6) V2a-D neurons, organized into four bins based on frequency. Dashed line indicates transition between slow carangiform and fast anguilliform modes. Bottom panel: as above, but for fast (FB, *n* = 11) and slow (SB, *n* = 5) V2a-B neurons. (**H**) Top panel: Box plots of max EPSC% for fast and slow V2a-D neurons from panel G, collapsed into single slow (<35 Hz) and fast (>35 Hz) bins for purposes of statistical analysis (Wilcoxon signed-rank test; FD, *V* = 0, p < 0.01, n = 9; SD, *V* = 21, p < 0.05, n = 6). Bottom panel: As above but for V2a-B neurons (Wilcoxon signed-rank test; FB, *V* = 2, p < 0.01, n = 11; SB, *V* = 15, p = 0.063, n = 5). Boxed plots are superimposed on trendlines from individual neurons. Note, one cell each for V2a-D and V2a-B slow subtypes behaved more like fast subtypes (dashed lines). (**I**) Top panel: box plots of the maximum IPSC per cycle as a percentage of the maximum current (max IPSC%) for fast (FD, *n* = 7) and slow (SD, *n* = 7) V2a-Ds organized as in panel G. Bottom panel: as

*Figure 5 continued on next page*

*Figure 5 continued*

above but for fast (FB, *n* = 12) and slow (SB, *n* = 6) V2a-B neurons. (**J**) Top panel: box plots of max IPSC% for fast and slow V2a-D neurons from panel *I*, collapsed into single slow (<35 Hz) and fast (>35 Hz) bins for purposes of for statistical analysis (Wilcoxon signed-rank test; FD, $V = 0$, $p < 0.05$, $n = 7$; SD, $V = 1$, $p < 0.05$, $n = 7$). Bottom panel: as above, but for fast and slow V2a-B neurons (Wilcoxon signed-rank test; FB, $V = 0$, $p < 0.001$, $n = 12$; SB, $V = 4$, $p = 0.219$, $n = 6$). (**K**) Box plots of maximum excitatory and inhibitory current for FD (Wilcoxon signed-rank test; $V = 0$, $p < 0.01$, $n = 9$ for excitation, $n = 8$ for inhibition) and SD (Wilcoxon signed-rank test; $V = 0$, $p < 0.05$, $n = 8$ for excitation, $n = 7$ for inhibition) neurons. (**L**) As in *K* but for FB (Wilcoxon signed-rank test; $V = 1$, $p < 0.001$, $n = 16$ for excitation, $n = 15$ for inhibition) and SB (Wilcoxon signed-rank test; $V = 0$, $p < 0.05$, $n = 6$ for excitation, $n = 6$ for inhibition) neurons. Statistically significant differences are denoted as follows: *$p < 0.05$; **$p < 0.01$; ***$p < 0.001$.

same pattern for fast V2a neurons (*Figure 5I*, *left*), significantly increasing at fast speeds (*Figure 5J*). For slow V2a-D neurons excitation was significantly lower at higher frequencies (*Figure 5G*, *top right*; *Figure 5H*, *top*) and inhibition was significantly higher (*Figure 5I*, *top right*; *Figure 5J*, *top*). However, for slow V2a-B neurons neither excitation (*Figure 5G*, *bottom right*; *Figure 5H*, *bottom*) nor inhibition (*Figure 5I*, *bottom right*; *Figure 5J*, *bottom*) changed significantly when pooled, despite the decreases in excitation (*n* = 4 of 6) and increases in inhibition (*n* = 5 of 6) in the majority of neurons. These patterns of excitation and inhibition are consistent with recruitment patterns, where slow V2a-D neurons are silenced at fast speeds and slow V2a-B neurons are recruited albeit with lower reliability (*Figure 4F*).

While the increase in inhibition at fast speeds to slow V2a neurons makes sense in terms of their disengagement during fast anguilliform mode (*McLean et al., 2008*), the increase in inhibition to fast V2a neurons clearly does not have a similar impact, despite being normalized to excitability. Indeed, when we compared the maximum excitation and inhibition to V2a-Ds (*Figure 5K*) and V2a-Bs (*Figure 5L*), maximum inhibition was orders of magnitude larger than maximum excitation. Even when controlling for differences in driving forces experienced at resting potentials, it would be difficult for V2a neurons to fire if excitation and inhibition arrived concurrently.

Thus, measurements of maximum levels of excitatory and inhibitory drive that include fast phasic and slower tonic components are consistent with recruitment patterns. These data also suggest the timing of inhibition plays a critical role in shaping the rhythmic output of V2a neurons more broadly.

## Differences in the timing of synaptic drive better explain rhythmic V2a recruitment

To compare the timing of compound EPSCs and IPSCs we normalized the amplitude to the maximum (1) and minimum (0) values for that cycle and normalized the timing to the phase of the swim cycle, with the centers of two successive motor bursts demarcating phases 0 and 1 (*Figure 6A*). To simplify comparisons between peaks in excitation and troughs in inhibition, we inverted the sign of excitation. By convention, synaptic currents arriving between 0.75 and 0.25 are described as 'in-phase' with local motor output, while currents arriving 0.25 and 0.75 are described as 'anti-phase'. In frog tadpoles and adult lampreys, in-phase currents are assumed to arise from recurrent sources on the ipsilateral side, while anti-phase currents arise from the reciprocal sources on the contralateral side (*Soffe and Roberts, 1982*; *Russell and Wallén, 1983*). This means that differences in the timing of peaks of excitation and inhibition can be used to draw inferences about patterns of connectivity along and across the body, with phase-advanced inputs from more rostral regions and phase-delayed inputs from more caudal ones.

First, to assess differences in the timing of compound EPSCs and IPSCs, we low pass filtered the raw current trace to identify peaks in- and anti-phase (*Figure 6A*, *black dots*), which were then plotted on a circular axis and compared to cell-attached spiking activity (*Figure 6B–D*). For all V2a neurons, spikes were exclusively distributed in-phase with local motor output (*Figure 6B*), with largely overlapping distributions that were broader for fast V2a-Ds (*Figure 6B*, *top*) and slow V2a-Bs (*Figure 6B*, *bottom*), consistent with spike distributions observed in real time (*Figure 4D*). However, there were type-specific differences in the distributions of peak excitation and inhibition.

For both fast and slow V2a-Ds phasic excitation began early in anti-phase, while the distribution of phasic excitation in fast and slow V2a-Bs was more coherent and trailed that of V2a-Ds (*Figure 6C*). This is consistent with the proposed hierarchical organization of V2a neurons, with V2a-Ds occupying a position upstream of V2a-Bs, which is not as obvious based purely on spike distributions (*Figures 4D and 6B*). For fast V2a-Ds, phasic inhibition was also broadly distributed, with a dip that coincided with

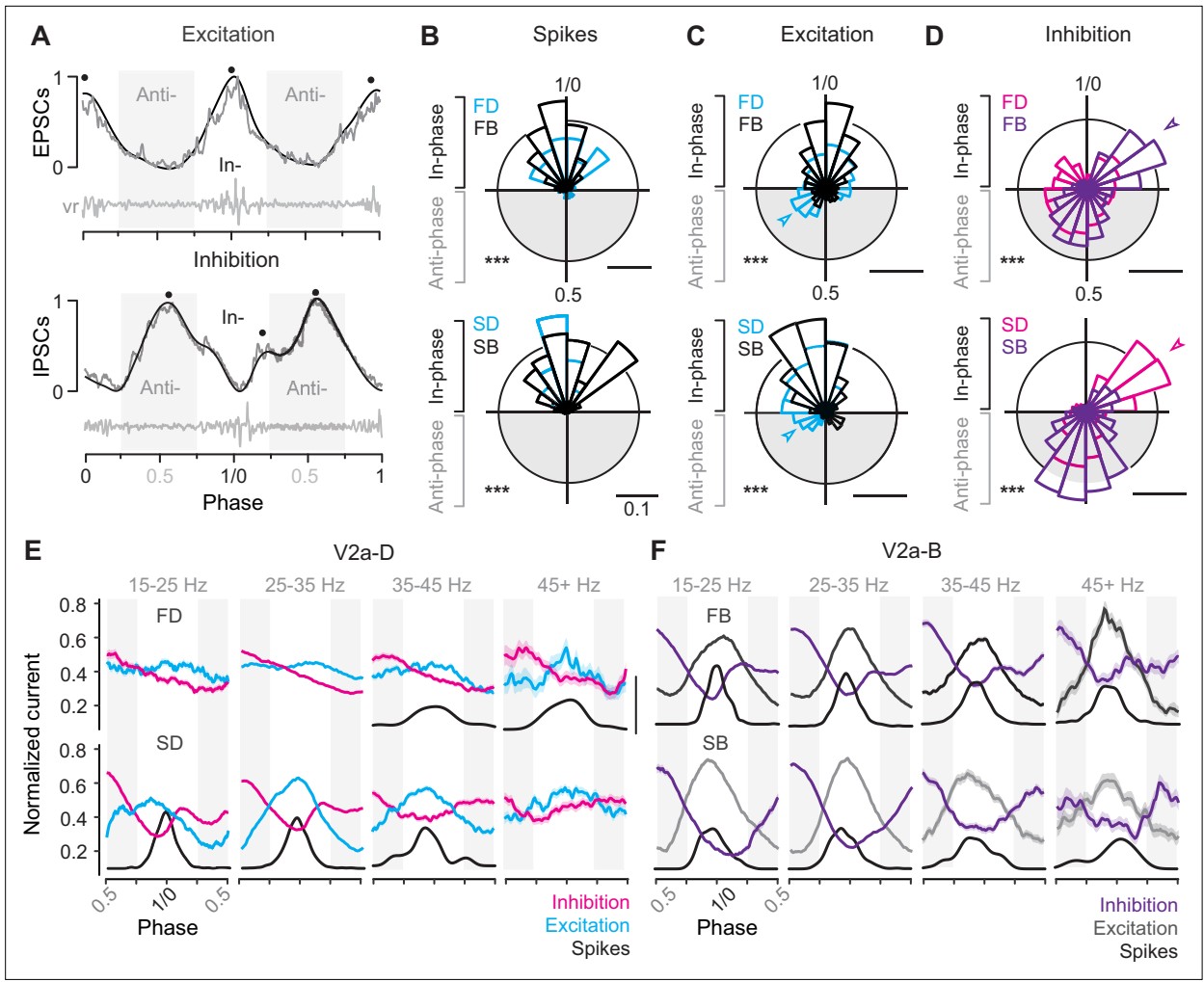

**Figure 6.** Differences in timing of synaptic drive related to speed and V2a cell type. (**A**) Raw (gray) and low-pass filtered (black) traces of post-synaptic current (PSC), normalized to the peak (1) and trough (0) of phasic excitation (left) and inhibition (right). Corresponding ventral root (vr) bursts used to define in- and anti-phase (gray shaded boxes) currents are shown below. Block dots denote the peaks identified by the findpeaks() function in MATLAB. Note, excitation has been inverted to simplify comparisons to inhibition. (**B**) Top panel: circular bar plots of spike phase for fast descending (FD, $n$ = 9) and bifurcating (FB, $n$ = 16) V2a neurons (Watson's two-sample test of homogeneity; $F$ = 0.483, p < 0.001). Bottom panel: circular bar plots of spike phase for slow descending (SD, $n$ = 8) and bifurcating (SB, $n$ = 6) neurons (Watson's two-sample test of homogeneity; $F$ = 3.711, p < 0.001). Scale bars, 10% total distribution. (**C**) Top panel: circular bar plots of phase normalized peaks in low pass filtered excitatory post-synaptic currents (EPSCs) for fast descending (FD, $n$ = 9) and fast bifurcating (FB, $n$ = 11) V2a neurons (Watson's two-sample test of homogeneity; $F$ = 5.941, p < 0.001). Bottom panel: circular bar plots of phase normalized peaks in low pass filtered EPSCs for slow descending (SD, $n$ = 7) and slow bifurcating (SB, $n$ = 6) neurons (Watson's two-sample test of homogeneity; $F$ = 2.548, p < 0.001). Scale bars, 10% total distribution. (**D**) Top panel: circular bar plots of phase normalized peaks in low pass filtered inhibitory post-synaptic currents (IPSCs) for fast descending (FD, $n$ = 7) and fast bifurcating (FB, $n$ = 12) neurons (Watson's two-sample test of homogeneity; $F$ = 3.646, p < 0.001). Arrowhead indicates prominent in-phase IPSC component for fast V2a-Bs. Bottom panel: circular bar plots of phase normalized peaks in low pass filtered IPSCs for slow descending (SD, $n$ = 7) and slow bifurcating (SB, $n$ = 6) neurons (Watson's two-sample test of homogeneity; $F$ = 5.426, p < 0.001). Arrowhead indicates prominent in-phase IPSC component for slow V2a-Ds. Scale bars, 10% total distribution. (**E**) Line plots from V2a-D neurons of averaged excitation (Ex; FD, $n$ = 9; SD, $n$ = 6), averaged inhibition (In; FD, $n$ = 7; SD, $n$ = 7), and spike densities (S; FD, $n$ = 7; SD, $n$ = 7) broken into four ventral root burst frequency (Hz) bins. Post-synaptic current was normalized on a cycle-by-cycle basis, such that the minimum and maximum current per cycle equaled 0 and 1, respectively. Scale bar for spike densities, 50% total distribution. (**F**) As in panel $E$, but averaged excitation (Ex; FB, $n$ = 11; SB, $n$ = 5), inhibition (In; FB, $n$ = 12; SB, $n$ = 6), and spike densities (S; FB, $n$ = 16; SB, $n$ = 6) for V2a-B neurons.

the peak distributions of both excitation and spiking activity (*Figure 6D*, *top*). For slow V2a-Ds, there were two clear peaks in phasic inhibition, one immediately trailing in-phase excitation (*Figure 6D*, *bottom arrow*) and one peaking anti-phase. Remarkably, for V2a-B neurons, we observed the opposite pattern for phasic inhibition. Fast V2a-Bs received peaks in- and anti-phase (*Figure 6D*, *top arrow*),

while slow V2a-Bs received predominantly anti-phase inhibition (*Figure 6D*, *bottom*). Notably, the presence of in-phase inhibition was also associated with a more coherent distribution of spikes in slow V2a-Ds and fast V2a-Bs (cf., *Figure 6B*).

Next, to get a better sense of how patterns of phasic excitation and inhibition ultimately shape phasic firing patterns within different types at different speeds, we binned the data by frequency as above and examined averages of normalized raw current traces. Among the V2a-Ds, fast subtypes exhibited reciprocal inhibition at all frequencies, with phasic peaks in excitation and inhibition most obvious at higher frequencies coincident with spiking activity (*Figure 6E*, *top*). Among slow V2a-Ds, coherent peaks in reciprocal and recurrent inhibition were most obvious at slow speeds coincident with phasic excitation and spiking behavior (*Figure 6E*, *bottom*). Among the V2a-Bs, patterns of reciprocal and recurrent inhibition were consistent across all speed bins, with peaks in phasic excitation and dips in phasic inhibition coincident with spiking activity (*Figure 6F*).

Thus, despite differences in cellular rhythmogenic capacity, these observations suggest all types of V2a neurons rely on phasic excitation and inhibition to sculpt their rhythmic firing behavior during swimming. Peaks in phasic excitation arrive earlier among V2a-D neurons consistent with their hierarchical positioning upstream from V2a-B neurons, and the relative coherence of phasic excitatory and inhibitory drive at different frequencies is consistent with speed modularity in this population. Among V2a-B neurons, the coherence of phasic excitatory and inhibitory drive over the entire frequency range is consistent with a role integrating inputs from the different speed modules. Critically, these data also suggest reciprocal inhibition plays a key role in rhythmogenesis at fast speeds, while recurrent inhibition plays a more dominant role in rhythmogenesis at slow speeds.

## Synaptic silencing of reciprocal inhibition has speed-dependent effects on rhythm versus pattern

Our data thus far suggest that reciprocal inhibition plays a key role in rhythmogenesis at fast speeds via V2a-D neurons, while it plays a more dominant role patterning output at slow speeds via V2a-B neurons. To test this idea, we used a genetic approach to synaptically silence commissural, glycinergic Dmrt3a-labeled dI6 neurons (*Figure 7A, B*), which participate in swimming over a range of speeds in larvae (*Kishore et al., 2020*; *Satou et al., 2020*). Specifically, we crossed compound Tg[Dmrt3a:Gal4;UAS:pTagRFP] fish (*Satou et al., 2013*; *Kishore et al., 2020*) with Tg[UAS:BoTxBLC-GFP] fish (*Sternberg et al., 2016*), to express botulinum toxin (light chain) fused with GFP (BoTx-GFP) selectively in dI6 neurons and monitored the impact on locomotor rhythmogenesis using contralateral ventral rootlet recordings. As controls, we performed recordings from heterozygous sibling larvae that lacked expression of BoTx-GFP.

First, to confirm the efficacy of synaptic silencing, we analyzed the distribution of dI6 neurons expressing GFP-tagged BoTx compared to the entire population of BoTx-negative dI6 neurons labeled by pTagRFP (*Figure 7C*). BoTx-GFP-labeled neurons were observed throughout the DV range of the dI6 population (*Figure 7C*, *left and middle*), with a significantly higher proportion observed more ventrally (*Figure 7C*, middle and *right*). Since dI6 neurons located more ventrally participate in faster swimming movements (*Kishore et al., 2020*), we expected the impact of synaptic silencing would be more obvious at higher frequencies.

Consistent with this idea, in dI6 BoTx-GFP fish we frequently observed synchronous bilateral activity at the beginning of swim episodes when frequency is typically highest (cf., *Figure 1G*), which could transition to rhythmic, alternating activity nearer the end of the episode (*Figure 7D*). In control larvae, left–right phase values in the first and subsequent swim episodes had peak distributions that rarely deviated from 0.5, as you would expect for perfect alternation during locomotor rhythmogenesis (*Figure 7E*, *left*). On the other hand, for BoTx-GFP fish, left–right phase values were more broadly distributed, especially at the beginning of the episode, which was more obvious in the initial swim episode compared to subsequent ones (*Figure 7E*, *right*). This is also consistent with differences in peak frequencies observed near the beginning of swim episodes, where they are always higher in the episode immediately following the stimulation (cf., *Figure 1G*).

The lack of coordinated rhythmogenic output across the body at fast speeds was also apparent when comparing ventral rootlet burst frequencies. In dI6-GFP fish, higher-frequency activity overlapping higher-frequency activity in controls was observed at the beginning of swim episodes (*Figure 7D*, black dots), however this was rarely associated with higher-frequency activity in the corresponding

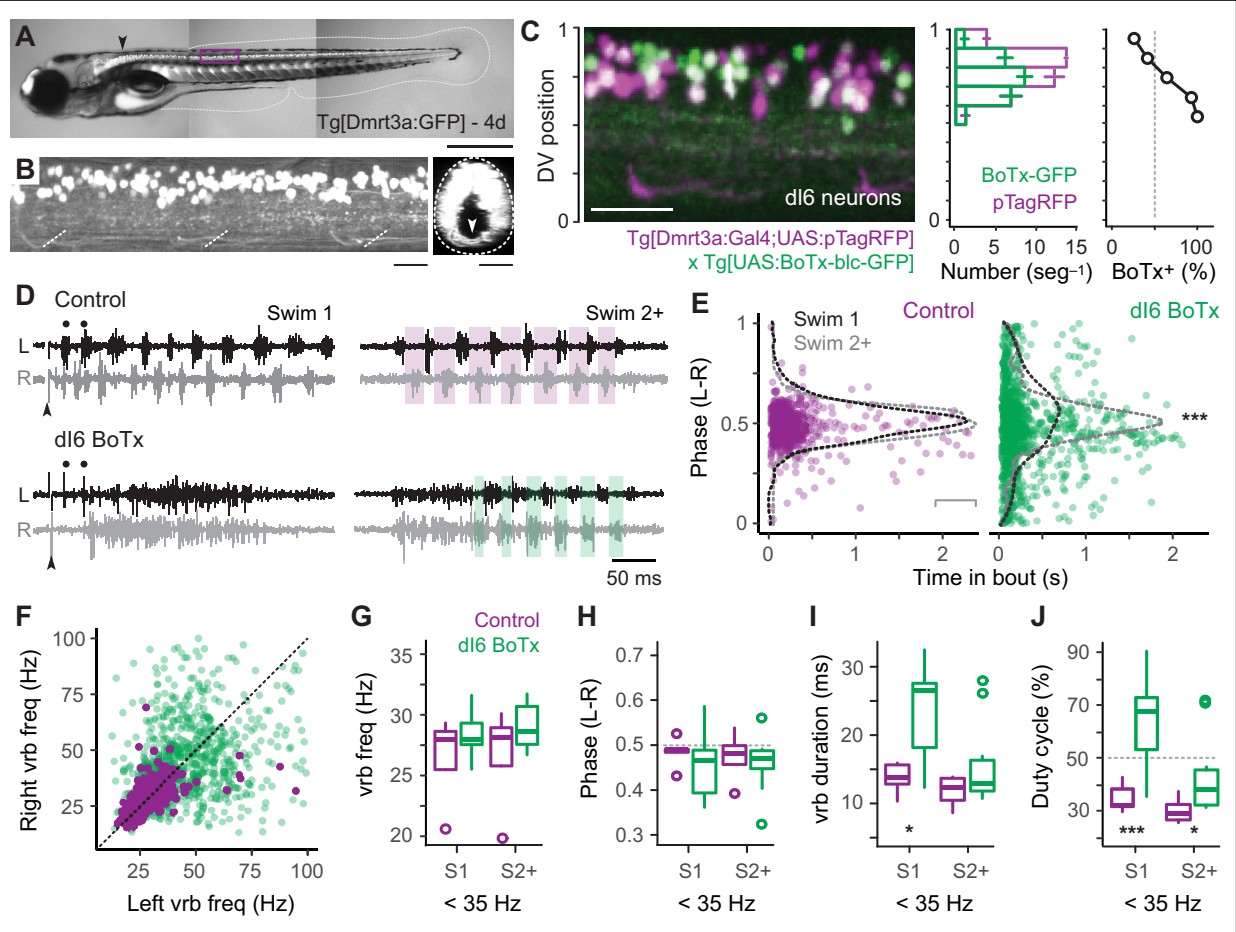

**Figure 7.** Speed-dependent impact of silencing dI6 neurons on rhythm versus pattern. (**A**) Composite DIC/epifluorescence image of Tg[Dmrt3a:GFP] larval zebrafish at 4 days post fertilization (4d). Dorsal is up and rostral is left. Scale bar, 0.5 mm. (**B**) Left panel: confocal image of the Tg[Dmrt3a:GFP] spinal cord from segments 10–12 at midbody (purple box in *A*). Scale bar, 20 μm. Right panel: coronal view of confocal image to the left illustrating commissural axons of dI6 neurons (white arrowhead). Dashed lines indicate boundaries of spinal cord. Scale bar, 20 μm. (**C**) Left panel: composite confocal image of all dI6 somata labeled with pTagRFP (purple) and subsets of dI6 somata labeled with botulinum toxin-tagged GFP (green), using a combination of compound transgenic lines (noted below). Boundaries of spinal cord are normalized to 0–1 in the dorsoventral (DV) axis. Scale bar, 30 μm. Middle panel: bar plots of average numbers (± SEM) of pTagRFP-labeled dI6 neurons (purple) and BoTx-GFP-labeled dI6 neurons (green) per midbody segment (*n* = 5–6 segments from 3 fish) along the DV axis. The DV distributions of pTagRFP expressing (*n* = 517) and BoTx-GFP expressing dI6 neurons (*n* = 842) are significantly different (Wilcoxon rank-sum test; *W* = 184,452, p < 0.001). Right panel: line plot of percentage of total dI6 neurons labeled with BoTx-GFP along the DV axis. Dashed gray line indicates 50% of the distribution. (**D**) Top panels: examples of extracellular ventral rootlet recordings recorded on the left (L) and right (R) sides in sibling control larvae lacking BoTx-GFP expression. Two consecutive swim episodes are illustrated, the first (Swim 1) is evoked by a mild electrical stimulus (s) to the tail fin (at arrowhead, artifact blanked), with subsequent episodes occurring intermittently after the first (Swim 2+). Scale bar, 500 ms. Bottom panels: examples from BoTx-GFP fish with coincident bilateral activity. Black dots illustrate ipsilateral bursts that match frequencies observed in controls. Shaded purple and green boxes illustrate left–right alternation. (**E**) Left panel: scatter plot of left–right (L–R) phase versus the time in the swim episode (s) in control siblings. Superimposed on pooled data are density plots from the swimming episode immediately following the stimulus (Swim 1) and subsequent swim episodes (Swim 2+), which are not significantly different following a two-sample Kolmogorov–Smirnov test (*D* = 0.084, p = 0.305, *n* = 330 cycles for Swim 1 from 5 fish and 219 cycles for Swim 2+ from 4 fish). Right panel: scatter plot of BoTxBLC-GFP fish illustrating collapse of left–right alternation and rhythmicity primarily at the beginning of the swimming episodes. In this case, density plots for Swim 1 versus Swim 2+ are significantly different (***, two-sample Kolmogorov–Smirnov test; *D* = 0.163, p < 0.001, *n* = 428 cycles for Swim 1 from 11 fish and 540 cycles for Swim 2+ from 10 fish). (**F**) Scatter plot analyzing ventral root burstlet (vrb) frequency for control (purple) and BoTx-GFP fish (green) on the left and right sides on a cycle-by-cycle basis. (**G**) Box plots of ventral rootlet burstlet frequency for control and BoTx-GFP fish during carangiform swimming below 35 Hz for Swim 1 (Wilcoxon rank-sum test; *W* = 35, p = 0.454, *n* = 11 BoTx-GFP and 5 control fish) and Swim 2+ (*W* = 26, p = 0.453, *n* = 10 BoTx-GFP and 4 control fish). (**H**) Box plots of ventral rootlet burstlet left–right phase for control and BoTx-GFP fish during carangiform swimming below 35 Hz for Swim 1 (Wilcoxon rank-sum test; *W* = 18, p = 0.439, *n* = 11 BoTx-GFP and 5 control fish) and Swim 2+ (*W* = 17, p = 0.733, *n* = 10 BoTx-GFP and 4 control fish). (**I**) Box plots of ventral rootlet burstlet duration for control and BoTx-GFP fish during carangiform swimming below 35 Hz for Swim 1 (Wilcoxon rank-sum test; *, *W* = 48, p < 0.05, *n* = 11 BoTx-GFP and 5 control fish) and Swim 2+ (*W* = 27, p = 0.374, *n* = 10 BoTx-GFP and 4 control fish). (**J**) Box plots of ventral rootlet burstlet duty cycle for control and BoTx-GFP fish during carangiform swimming below

*Figure 7 continued on next page*

*Figure 7 continued*

35 Hz for Swim 1 (Wilcoxon rank-sum test; **, $W = 52$, $p < 0.01$, $n = 11$ BoTx-GFP and 5 control fish) and Swim 2+ (*, $W = 27$, $p < 0.05$, $n = 10$ BoTx-GFP and 4 control fish). Statistically significant differences are denoted as follows: *$p < 0.05$; **$p < 0.01$; ***$p < 0.001$.

cycle on the other side (*Figure 7F*). In controls, cyclical frequencies on the left and right sides were symmetrical, with most of the datapoints falling along unity (*Figure 7F*). Spinal motor neurons exhibit a range of rhythmogenic cellular properties themselves (*Menelaou and McLean, 2012*), which likely contribute to the higher-frequency bursts observed unilaterally. Thus, the disorganized unilateral bursting during bilaterally synchronous output at the onset of swimming in dI6 BoTx-GFP fish is consistent with the selective disruption of rhythmogenesis at fast speeds.

Finally, we turned our attention to lower-frequency swimming observed nearer the end of swimming, which was less impacted by BoTx expression. For purposes of analysis, we divided the dataset into the initial (Swim 1) and subsequent (Swim 2+) swim episodes, and focused on ventral rootlet burst frequencies below 35 Hz. A comparison of burst frequency and phase revealed no significant differences between dI6 BoTx-GFP and control fish, consistent with a lack of impact on timing and rhythm generation at low frequencies (*Figure 7G, H*). However, we did observe an increase in burst duration (*Figure 7I*), which increased the percentage of the swim cycle occupied by motor output, or duty cycle (*Figure 7J*). These effects on were most obvious in the initial swim episode, where duty cycles could reach 90% (*Figure 7J*). Normally, duty cycles must not exceed 50% of the cycle period to avoid synchronous bilateral contractions during swimming (*Grillner, 1974*). Thus, at slow speeds the dominant impact of synaptic silencing of dI6 neurons is on motor output, not rhythmogenesis. Collectively, these observations are consistent with the idea that reciprocal inhibition plays a more dominant role in rhythm generation at fast speeds and a more dominant role in pattern formation at slow speeds.

## Discussion

Our work was motivated by the opaque origins of spinal rhythmicity at different locomotor speeds and the potential for zebrafish to clarify this fundamental issue. We focused on Chx10-labeled spinal V2a neurons given their conserved role in locomotion in zebrafish and mice (*McLean and Dougherty, 2015*). In mice, increases in speed involve graded increases in the cyclical frequency of rhythmic activity within limbs and abrupt changes in the coordination between them that define different gaits, like walking, trotting, and bounding (*Bellardita and Kiehn, 2015*). In larval zebrafish, there are two 'gaits' during forward swimming as defined by limb coordination – slow carangiform mode which involves cyclical abduction/adduction of pectoral fins that alternates across the body and fast anguilliform mode which involves bilateral fin adduction (*Thorsen et al., 2004*). The transition in fin coordination occurs abruptly between tail beat frequencies of 30–40 Hz in larvae, with graded increases in the cyclical frequency and intensity of body bends leading to increased yaw displacement of the head at faster speeds (*Thorsen et al., 2004*; *McLean et al., 2008*).

Earlier studies of larval zebrafish found that V2a neurons active at slow speeds are inhibited as V2a neurons active at fast speeds are recruited (*McLean et al., 2008*). The idea that different gears or modules are responsible for different speeds of locomotion is easier to understand in the context of different gaits. For swimming, faster speeds require not only higher-frequency oscillations, but also larger amplitude body bends that propagate more quickly to generate reactive thrust against the water column for effective propulsion (*Grillner, 1974*). Among descending V2a neurons, those recruited at faster speeds have longer intersegmental axons (*Menelaou et al., 2014*) and form stronger connections with other V2a neurons (*Menelaou and McLean, 2019*). Among bifurcating V2a neurons, intersegmental axon projection distances are more homogeneous, membrane time constants and axon conduction velocities are faster, and connections to motor neurons are stronger and more reliable (*Menelaou et al., 2014*; *Menelaou and McLean, 2019*; *Menelaou et al., 2022*). Thus, as V2a-D neurons with longer axons and V2a-B neurons with stronger premotor connections are engaged at higher frequencies to achieve higher amplitude, faster propagating body bends, V2a neurons whose segmental output is temporally incompatible with fast whole-body coordination are disengaged.

While modular organization makes sense in terms of coordination, the origin of rhythmicity at different speeds was assumed to be the same, whether via cellular properties or synaptic drive or some combination of the two (*Marder and Calabrese, 1996*; *Marder and Bucher, 2001*). From

our recordings of spinal motor neurons (*Menelaou and McLean, 2012*; *Kishore et al., 2014*), we were expecting that cellular properties are more important for rhythmogenesis at slow speeds, while synaptic drive dominates at fast speeds, which could also vary based on V2a type. Our recordings revealed that all V2a neurons demonstrate some form of rhythmicity in response to current steps. The idea that different types of V2a interneurons exhibit different firing properties is not unexpected (*Dougherty and Kiehn, 2010*; *Zhong et al., 2010*; *Menelaou and McLean, 2019*; *Song et al., 2020*). However, thanks to our ability to link cellular properties to natural recruitment patterns we reveal V2a-Ds alone would qualify as 'pacemakers' (*Marder and Calabrese, 1996*; *Marder and Bucher, 2001*), with speed-dependent differences in intrinsic firing mode within V2a-Ds. Specifically, V2a-D neurons recruited at swim frequencies below ~40 Hz during carangiform swimming generate intrinsic bursting at those frequencies, with higher-frequency spikes riding atop lower-frequency membrane oscillations. Intrinsic oscillations are more sensitive to levels of current injection than spike rates, so increases in tonic excitatory drive would lead to increased burst frequency and thus swim frequency. V2a-D neurons recruited at frequencies above ~40 Hz during anguilliform swimming generate tonic spiking at those frequencies, so increases in tonic excitatory drive would lead to increases in firing frequency and thus swim frequency. In V2a-B neurons, chattering and bursting properties in response to current steps do not reflect their natural firing patterns during swimming, and so we expected they would rely more on pacemaking inputs from V2a-Ds to shape their rhythmicity.

Surprisingly, however, despite clear differences in cellular properties appropriate to generate swimming rhythms, phasic firing in both V2a-Ds and V2a-Bs is best explained by phasic patterns of synaptic excitation and inhibition. Since phasic excitatory input is not tonic and already rhythmic by the time it arrives, this suggests rhythms are generated by more rostral V2a-Ds. Optogenetic stimulation of V2a neurons and spinal lesion studies support a rostral bias in spinal rhythmicity in larval zebrafish (*Wiggin et al., 2012*; *Kimura et al., 2013*), consistent with studies of spinal locomotor networks in lampreys and tadpoles, where neurons nearer the hindbrain/spinal cord boundary have a higher capacity for rhythmogenesis (*Li et al., 2010*; *Buchanan, 2018*). Our synaptic silencing of dI6 neurons also supports this idea. Previous work using diphtheria toxin to kill dI6 neurons employed an elegant Hox-based strategy to limit expression to the spinal cord (*Satou et al., 2020*). In contrast to our observations, zebrafish larvae with ablated spinal dI6 neurons can still generate fast swimming. Notwithstanding differences in cellular ablation versus synaptic silencing approaches and their relative sensitivity to potential compensatory mechanisms (*Schnerwitzki et al., 2018*; *Del Pozo et al., 2020*), our strategy also included dI6 neurons that extended into the hindbrain (*Figure 7A*, *arrow*). By subtractive reasoning, this argues that reciprocal inhibition nearer the hindbrain/spinal cord is essential for generating high-frequency rhythms.

Regardless, while all midbody V2a neurons rely on synaptic drive to fire rhythmically at all speeds, the patterns of synaptic excitation and inhibition responsible for rhythmogenesis varies in a cell-type- and

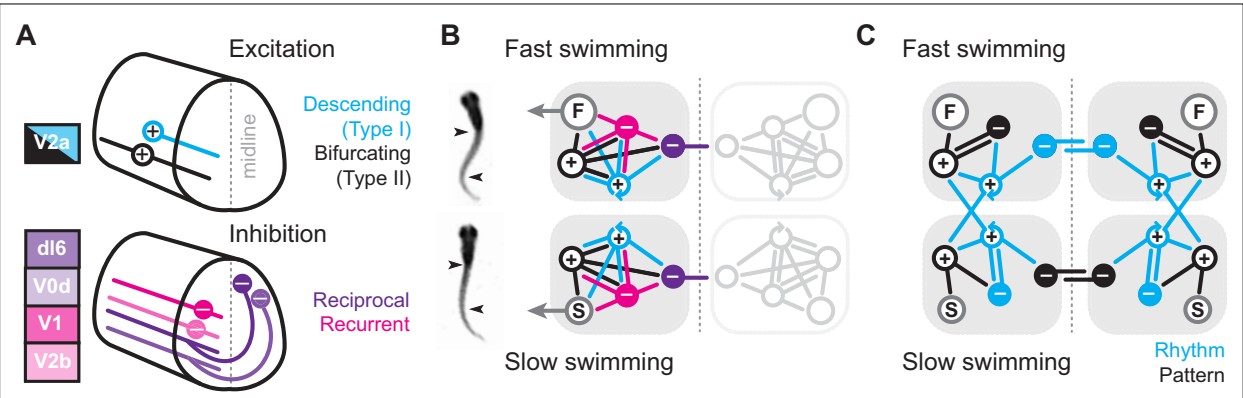

**Figure 8.** Summary diagrams of cell-type-specific origins of locomotor rhythmicity. (**A**) Schematic illustrating the major spinal interneuron types that provide excitation (top) and inhibition (bottom) during locomotion. Dorsal is up and rostral is left. See Discussion for details. (**B**) Schematic illustrating the potential wiring diagrams between major spinal interneuron types (+, excitatory; −, inhibitory) for fast swimming (top) and slow swimming (bottom). Snapshots from high-speed videos are illustrated to the left for fast anguilliform and slow carangiform swimming. F, fast motor pools; S, slow motor pools. Dashed vertical line is midline. Reciprocal populations are shaded gray for simplicity. (**C**) Schematic focusing on the major connections that differentiate fast and slow swimming circuits. Rhythm-generating circuitry is in blue and pattern-forming circuitry is in black.

speed-dependent manner. Our results combined with prior work allow us to put together a potential model explaining the cellular and synaptic origins of rhythmogenesis at different speeds of swimming. In zebrafish and mice (*Hayashi et al., 2018*; *Menelaou and McLean, 2019*), V2a neurons have been categorized into descending (I) and bifurcating (II) subtypes (*Figure 8A*, *top*). Our data only strengthen cross-species links by demonstrating that type I and II V2a neurons in larval zebrafish not only differ in axon trajectories, rostro-caudal distribution, size, Chx10 levels, and connectivity, but also in their ML distribution in the spinal cord (*Figure 1C*).

Since both V2a subtypes project ipsilaterally, recurrent (or feedback) inhibition must originate from ipsilateral sources, while reciprocal (or feedforward) inhibition originates from commissural sources. In zebrafish, as in mice, there are at least two potential sources of ipsilateral inhibition, V1 neurons (*Higashijima et al., 2004*) and V2b neurons (*Callahan et al., 2019*), and two potential sources of commissural inhibition (*Satou et al., 2020*), dI6 neurons and V0d neurons (*Figure 8A*, *bottom*). In our model, we include the two specific types of V2a neurons and single generic sources of recurrent and reciprocal inhibition, each of which has the potential to form connections with the other types (*Sengupta and Bagnall, 2023*). We do not include conserved sources of commissural excitation here (e.g., V0v and V3 neurons), but these will be important to include in any future models exploring intersegmental coordination and signal amplification (*Böhm et al., 2022*; *Kawano et al., 2022*; *Wiggin et al., 2022*). Notably, the operational definition of reciprocal versus recurrent inhibition would be flipped for commissural excitatory populations, with feedforward inhibition via ipsilateral interneurons and feedback inhibition via commissural interneurons.

The simplest scenario is that the wiring diagrams for slow and fast modules are mirror duplications distinguished only by the inclusion of slow or fast motor pools (*Figure 8B*). Consistent with this idea, our cell-attached and voltage-clamp data support distinct slow and fast type I V2a-Ds defined by selective within module connectivity. These neurons in turn converge on slow and fast type II V2a-Bs, which relay integrated frequency information to their respective motor pools (*Figure 8C*). In the fast module, rhythmogenesis orchestrated by type I V2a neurons relies on reciprocal inhibition. This is consistent with cell-type-specific optical ablation and silencing studies in adult lampreys (*Buchanan and McPherson, 1995*) and embryonic frogs (*Moult et al., 2013*) that suggest reciprocal inhibition is essential for rhythmogenesis during anguilliform swimming. Rhythmic activity can also be generated in surgically isolated hemi-cords and surgically divided spinal cords of lampreys and frog embryos (*Kahn and Roberts, 1982*; *Soffe, 1989*; *Cangiano and Grillner, 2003*; *Cangiano and Grillner, 2005*; *Li et al., 2010*). However, the functional relevance of these rhythms and the impact of lesion-induced plasticity on excitability and connectivity are still debated (*Cangiano et al., 2012*; *Messina et al., 2017*; *Parker, 2017*; *Cangiano and Grillner, 2018*; *McClellan, 2018*).

In the fast module, recurrent inhibition helps limit motor output via type II V2a-Bs. Type II V2a-Bs have relatively strong connections to motor neurons (*Menelaou and McLean, 2019*), so limiting their firing would ensure that motor output does not exceed 50% of the duty cycle and risk bilateral co-contraction as the interval between cyclical bursts decreases with increased speed (*Grillner, 1974*). On the other hand, the cell-type-dependent dominance of reciprocal versus recurrent inhibition is flipped when considering the slow module. Instead, recurrent inhibition is more obvious in type I V2a-D neurons, while type II V2a-B neurons rely exclusively on reciprocal inhibition to pattern their activity (*Figure 8C*). Until now recurrent inhibition was linked to limiting motor output (*Hultborn et al., 1979*), not to rhythmogenesis (see, however, *Lindén and Berg, 2021*).

It remains to be seen why reciprocal inhibition would be favored for rhythmogenesis at fast speeds and recurrent inhibition at slow speeds. One possibility is that feedforward inhibition is necessary to properly coordinate bilateral rhythmogenesis among type I V2a-D neurons with longer-range intersegmental connectivity. On the other hand, feedback inhibition among shorter-range type I V2a-D neurons provides an opportunity for local segmental rhythms that rely less on contralateral activity. Since slow type II V2a-B neurons lack feedback inhibition, their recruitment patterns rely exclusively on feedforward inhibition from the opposite side. This means stronger activity on one side should limit output on the other side without impacting timing and vice versa. In support, we recently explored the contribution of spinal V2a neurons to biased swimming during optomotor responses (*Jay et al., 2023*). We found that type I V2a-D neurons are direction-agnostic, but type II V2a-B neurons are direction-sensitive, with higher recruitment probabilities on the turning side associated with weaker motor output on the opposite side.

Interestingly, in contrast to low recruitment probabilities observed for type I V2a-D neurons during swimming evoked by somatosensory stimulation here (<50%; *Figure 5E, F*), for swimming during optomotor responses V2a-D neurons are recruited with much higher probabilities (>75%). This is not speed related since swim frequencies during optomotor responses rarely exceeded 35 Hz. Notably, the phase distribution of V2a-D spikes during optomotor responses (Figure 5H in *Jay et al., 2023*) better aligns with the phase distribution of EPSCs reported here (*Figure 6C*, *bottom*), meaning that type I V2a-Ds fire well in advance of type II V2a-Bs during optomotor responses. Thus, it appears that during somatosensory-evoked swimming, where intensity and frequency are coupled over a broad range of speeds, type I and II V2a neurons fire more synchronously than during visually evoked swimming, where intensity and frequency are uncoupled over a more narrow speed range. While this highlights the modular nature of timing (rhythm) and intensity (pattern) control within the spinal V2a population, future experiments testing this idea will rely on identifying molecular markers that distinguish slow and fast type I and II V2a neurons. Fortunately, recent work has identified promising candidates (*Pallucchi et al., 2024*).

Another open question is what role differences in cellular properties play during rhythmogenesis. First, our recordings were performed at midbody. It could be that rhythmogenic bursting and spiking properties are a general feature of type I V2a-D neurons, so more rostral neurons have identical cellular properties and rely less on phasic synaptic drive for fast and slow rhythmogenesis. Future recordings from V2a neurons in rostral spinal cord should help resolve this issue, however our dI6 synaptic silencing experiments suggest that reciprocal inhibition will be critical for rostral rhythmogenesis, at least at fast speeds. On the other hand, bursting properties could serve as a source of amplification for synaptic inputs to ensure reliable recruitment of rhythmogenic neurons or provide redundant sources of rhythmogenesis in cases where synaptic drive fails or the spinal cord is damaged. In this sense, cellular properties and synaptic drive would provide synergistic roles amplifying and sharpening rhythmic signals as they propagate segmentally.

In juvenile/adult zebrafish, V2a and motor neurons have been grouped into slow, intermediate, fast, and escape modules based on patterns of recruitment and phasic synaptic drive during fictive swimming (*El Manira, 2023*). In larvae, we also observe recruitment patterns among V2a neurons that would qualify as slow, intermediate, fast and escape based on their criteria (cf., *Figure 2G*). However, we also find that recruitment probability varies based on morphological type, with V2a-D neurons always firing more sparsely than V2a-B neurons, regardless of speed. Thus, low firing probability recorded at slow speeds does not necessarily predict high firing probability at fast speeds. Collectively, these observations suggest that increases in speed in juveniles/adults and larvae are initially accomplished via more reliable V2a neuron recruitment, as also reported in mice (*Zhong et al., 2011*). This fits studies of motor neurons, where relatively low initial forces rely on recruitment (*Enoka and Duchateau, 2017*; *Jay et al., 2023*), while higher forces are achieved by increasing firing rates (*Bhatt et al., 2007*; *Enoka and Duchateau, 2017*), in addition to silencing more easily recruited neurons (*Smith et al., 1980*; *Kyriakatos et al., 2011*; *Menelaou and McLean, 2012*).

In sum, our findings suggest that cell-type-specific modes of spinal rhythmogenesis are responsible for different speeds, which means a common mechanism of rhythmogenesis is insufficient to explain the full operational range of locomotion. Experiments combining computational modeling with acute segmental perturbations of distinct interneuron types are now needed to explore the functional impact of stratified spinal locomotor circuits with cell-type-specific feedforward and feedback inhibitory motifs.

## Materials and methods
### Fish husbandry
Tg[Chx10:GFP] fish (ZFIN ID: ZDB-ALT-061204-2) were used to target V2a neurons in the spinal cord (*Kimura et al., 2006*), and compound Tg[Dmrt3a:Gal4;UAS:pTagRFP] fish (ZFIN IDs: ZDB-FISH-150901-3557; ZDB-TGCONSTRUCT-210826-11) crossed with Tg[UAS:BoTxBLC-GFP] fish (ZFIN ID: ZDB-FISH-160816-31) were used to express botulinum-toxin light chain fused with GFP in commissural inhibitory dI6 interneurons (*Satou et al., 2013*; *Sternberg et al., 2016*; *Kishore et al., 2020*). Fish were maintained at 28.5°C on a 14-hr light/10-hr dark cycle in an aquatics facility staffed by Northwestern University's Center for Comparative Medicine. All experiments were performed at room

temperature between 4 and 5 days post fertilization (dpf) when zebrafish larvae are free swimming. Because 4–5 dpf larvae have not yet developed secondary sexual characteristics, their sex cannot be easily determined. Therefore, experiments were performed on larvae of either sex. All experimental protocols conform to National Institutes of Health guidelines on animal experimentation and were approved by the Northwestern University Institutional Animal Care and Use Committee (IS00019359 and IS00019319).

## Electrophysiology and analysis

Electrophysiological recordings were acquired using a Multiclamp 700A amplifier (Molecular Devices), a Digidata series 1322A digitizer (Molecular Devices), and pClamp 10 software (Molecular Devices). The intrinsic firing patterns of V2a neurons were assessed by performing 500ms long current steps at 1–2x rheobase in current-clamp mode. To compare the intrinsic firing properties of V2a neurons to their recruitment profiles during 'fictive' escape swimming, we performed simultaneous whole-cell current-clamp recordings of V2a neurons and extracellular recordings of the peripheral ventral rootlet. Synaptic currents during fictive swimming were assessed in dual whole voltage-clamp and extracellular rootlet recordings. Swimming was evoked by applying a mild and brief electrical shock (2–10 V; 0.1–0.4 ms) using a bipolar tungsten electrode placed over the tail fin.

For recordings larvae were first paralyzed using alpha-bungarotoxin (1 mg/ml in extracellular solution containing 134 mM NaCl, 2.9 mM KCl, 1.2 mM MgCl$_2$, 10 mM 2-(4-(2-hydroxyethyl)-1-piperazinyl)-ethanesulfonic acid (HEPES), 10 mM glucose, and 2.1 mM CaCl$_2$, adjusted to pH 7.8 with NaOH), immersed in extracellular solution containing anesthetic (MS-222), and secured left side up on a Sylgard lined glass bottom dish using custom etched tungsten pins through the notochord. To record the activity of the peripheral ventral rootlets, the skin over the muscle tissue at the midbody was removed using fine forceps. For access to spinal V2a neurons, the muscle tissue over one spinal segment was dissected away using a tool fashioned from sharpened tungsten wire. After the dissection, the preparation was rinsed and immersed in anesthetic-free extracellular solution. For cell-attached and whole-cell recordings, standard-wall (1 mm outer diameter) borosilicate capillaries (Sutter Instrument) were pulled to resistances between 5 and 16 MΩ using a micropipette puller (Flaming/Brown; Sutter Instrument). For peripheral motor nerve recordings, electrodes were made by pulling a patch electrode, breaking the tip, and then heat polishing and bending it using a microforge (MF-830; Narishige) to offset the approach angle for better contact between the electrode and muscle tissue. Peripheral motor nerve recordings were made no more than two segments caudal to the site of cellular recording.

For current-clamp recordings, patch electrodes were filled with intracellular solution containing (125–130 mM K-gluconate, 2–4 mM MgCl$_2$, 0.2–10 mM ethylene glycol-bis(β-aminoethyl ether)-N,N,N′,N′-tetraacetic acid (EGTA), 10 mM HEPES, 4 mM Na$_2$ATP, adjusted to pH 7.3 with KOH). For voltage-clamp recordings, the intracellular solution consisted of 122 mM CsMeSO$_3$, 0.1–1 mM QX314-Cl, 1 mM tetraethylammonium chloride (TEA-Cl), 3 mM MgCl$_2$, 10 mM HEPES, 1 mM EGTA, and 4 mM Na$_2$-ATP. CsMeSO$_3$ and QX-314 Cl were included in the solution to block voltage-dependent channels. Alexa Fluor 568 hydrazide (50 µmol/l) was also added to the intracellular solution for the post hoc confirmation of cell identity. Only neurons whose morphologies could be positively identified were included in this study.

V2a neurons were targeted on an epifluorescent microscope (Axio Examiner) under a ×40/1.0 NA water-immersion objective (W Plan-Apochromat; Zeiss) using a CCD camera (Rolera-XR; QImaging). The electrode was positioned using a motorized micromanipulator (MP-225; Sutter Instrument or Patchstar, Scientifica), while positive pressure was maintained using a pneumatic transducer (Model DPM-1B; Bio-Tek Instruments) to prevent the tip from clogging as the neuron was approached. The neuron was held at −65 mV during GΩ seal formation. Once the seal was achieved, for voltage-clamp experiments cell-attached recordings were performed to obtain the recruitment patterns of V2a neurons during fictive escape swimming before breaking into the neuron to record synaptic currents. A 5-mV hyperpolarizing pulse was then administered at a holding potential of −65 mV, and the steady-state current injected during the pulse was used to compute input resistance according to Ohm's law.

Excitatory currents were clamped at the reversal potential for inhibition calculated from our solutions, which is −75 mV and inhibitory currents were clamped at the reversal potential for excitation, which is +10 mV. These values were corrected for a calculated junction potential of −11 mV. Thus,

neurons were alternately held at the corrected potentials of −64 mV for EPSCs and 21 mV for IPSCs. To monitor series resistance throughout the experiment, recordings were alternately performed without and with 60% series resistance compensation. Standard corrections for bridge balance and electrode capacitance were applied for current-clamp recordings. Whole-cell electrical signals were filtered at 10 kHz and digitized at 100 kHz at a gain of 10 (feedback resistor, 500 MΩ). Extracellular signals from the peripheral motor nerves were amplified at a gain of 1000 and digitized with low- and high-frequency cutoffs set at 300 and 5000 Hz, respectively.

Whole-cell and extracellular recordings were annotated in Dataview to label individual spikes, ventral rootlet bursts, and swimming episodes. The onset and duration of spikes, ventral root bursts and swimming episodes were then batch exported into text files, which were then read into MATLAB (Mathworks) for further analysis. Instantaneous spike frequencies are the inverse of the period between two successive action potentials. The recruitment patterns of V2a neurons were quantified by computing the probability of observing spikes for all cycles present within each 5 Hz ventral root burst frequency bin from 10 to 65 Hz. The recruitment probability was then plotted as a function of ventral root burst frequency. The V2a neurons with an upward sloping recruitment reliability plot were classified as fast and those with a downward sloping plot were categorized as slow (cf., trendlines in *Figure 4E*).

Maximum excitatory and inhibitory currents reflect the maximum current received at any ventral root burst frequency. To compare frequency-dependent changes in synaptic currents across neuron types, they were normalized against the maximum synaptic current the neuron received. Normalized currents were then averaged in four discrete frequency bins spanning carangiform and anguilliform swimming modes (15–25, 25–35, 35–45, and 45+ Hz). To analyze the phasic timing of spiking and synaptic currents, they were normalized to the period of the corresponding cycle, such that the centers of successive motor bursts marked phases 0 and 1. Synaptic currents arriving between 0.75 and 0.25 were considered in-phase and those between 0.25 and 0.75 considered anti-phase. To analyze the relative timing of excitation and inhibition, we normalized amplitudes to the maximum current received in each cycle, such that the final values of normalized current ranged between 0 and 1. Currents were then averaged for all cycles present within the four frequency bins.

To quantify peaks in synaptic currents corresponding to the phasic inputs received during a swim cycle, current traces were low pass filtered at the cutoff frequency of 1000 Hz using the butterworth (second order) and zero phase digital filters in MATLAB. The latter was used to prevent the raw current values from undergoing a phase shift. Peaks were detected using the findpeaks() function in MATLAB. Counts of V2a spikes, EPSC's and IPSC's were plotted on a circular axis from phase 0 to 1. Data visualization and statistical analysis were performed in R.

To quantify left–right alternation in control and botulinum-toxin expressing dI6 neurons, dual ventral rootlet recordings were rectified and annotated in DataView. Owing to the highly irregular nature of the ventral rootlet activity in the dI6-silenced fish, automatic event detection parameters used to distinguish motor bursts in control conditions were unable to accurately demarcate the boundaries of motor bursts in BoTx-GFP fish. Therefore, all annotations for controls and BoTx-GFP were performed manually. Ventral root bursts at frequencies above a 100 Hz were excluded from analysis since this is the upper bound of their natural swimming range (*Müller and van Leeuwen, 2004*). Stimulus, burst, and bout onset and offset times for all recordings were batch exported into text files. Custom written MATLAB code was then used to classify Swim 1 and Swim 2+ episodes and identify corresponding bursts in the dual ventral rootlet recordings to compute phase values. Phase values were calculated from the center of the ventral rootlet motor burst.

## Imaging and analysis

After each electrophysiological recording, a series of epifluorescent and differential interference contrast images were captured using a CCD camera (Rolera-XR) and QCapture Suite imaging software (QImaging) to track the axonal trajectories of V2a neurons and confirm their morphology. Images were later analyzed in ImageJ. Confocal z-stacks of Tg[Chx10:GFP] and Tg[Dmrt3a:GFP] fish were captured on using a Zeiss LSM710 microscope using established protocols (*Menelaou et al., 2014*).

Soma size and DV locations of individually labeled V2a neurons were measured in ImageJ. DV and ML positions of V2a neurons in Tg[Chx10:GFP] were computed by obtaining the relevant positional coordinates from confocal images in Imaris (Oxford Instruments). Briefly, the spot detection

tool was used to obtain the coordinates of the centers of all somata and orthogonal slicers were used in conjunction with measurement points to extract the coordinates of the DV and lateral boundaries of the spinal cord as well as the central canal. The positions of the somata were then, respectively, normalized against the positions of the DV and ML boundaries to get the final DV and ML positions. V2a somata were classified into two groups based on the intensity of GFP expression. The GFP fluorescence intensity of V2a somata was exported from Imaris, and its density distribution was plotted in R. Because the density distribution of GFP fluorescence intensity was bimodal, the minimum between the two modes of the distribution was used as a cutoff to split V2a neurons into classes of low and high GFP expression. DV positions of dI6 neurons in compound Tg[Dmrt3a:Gal4;UAS:pTagRFP] × Tg[UAS:BoTxBLC-GFP] larvae were calculated using the relevant positional coordinates from confocal images in Imaris and normalizing them to the dorsal and ventral boundaries of spinal cord determined using differential contrast images.

## High-speed filming

To illustrate the differences in body shape and frequency with time, we include a video of a freely swimming 5-day-old larva captured at 2000 frames/s using a FASTCAM Mini AX50 (Photron), evoked using a sharpened tungsten pin to stimulate somatosensory afferents (*Video 1*).

## Statistics

All data were first tested for normality to select the appropriate statistical test. None of the data were normally distributed, so nonparametric tests were used to assess statistical significance. The difference between the density distributions of ML and DV somatic positions was tested using the two-sample Kolmogorov–Smirnov test. Correlations were measured using Spearman's rank correlation test. Comparisons between two groups of paired or unpaired data were performed using the Wilcoxon signed-rank test and the Wilcoxon rank-sum test, respectively. For data comprised of more than two groups, differences were assessed using the Kruskal–Wallis test, followed by pairwise comparisons using Dunn's test with a post hoc Benjamini–Hochberg false-discovery rate correction for multiple comparisons. For circular plots, Watson's two-sample test of homogeneity was used to determine whether the spike and post-synaptic current distributions in two conditions were significantly different from each other. Statistically significant differences are denoted as follows: $*p < 0.05$; $**p < 0.01$; $***p < 0.001$. Note that it was not always possible to collect cell-attached data and voltage-clamp recordings of excitation and inhibition in the same cells, so $n$ numbers reported in the text can vary between conditions. The same holds true for measured parameters in current-clamp recordings, where soma size, DV position, input resistance, rheobase, spike rates, and recruitment order in the same neurons were not always possible.

## Acknowledgements

We are extremely grateful to Katelyn Young for technical support. Financial support provided by NIH R21 NS125187 and NIH R21 NS125207.

## Additional information

### Funding

| Funder | Grant reference number | Author |
| --- | --- | --- |
| National Institutes of Health | R21 NS125187 | David L McLean |
| National Institutes of Health | R21 NS125207 | David L McLean |

The funders had no role in study design, data collection, and interpretation, or the decision to submit the work for publication.

## Author contributions
Moneeza A Agha, Data curation, Formal analysis, Investigation, Visualization, Writing – review and editing; Sandeep Kishore, Investigation; David L McLean, Conceptualization, Supervision, Funding acquisition, Visualization, Writing – original draft, Writing – review and editing

## Author ORCIDs
Moneeza A Agha ⬢ https://orcid.org/0000-0001-7868-8325
David L McLean ⬢ https://orcid.org/0000-0001-6337-2301

## Ethics
All experimental protocols conform to National Institutes of Health guidelines on animal experimentation and were approved by the Northwestern University Institutional Animal Care and Use Committee (Protocol numbers IS00019359 and IS00019319).

Reviewer #1 (Public review): https://doi.org/10.7554/eLife.94349.3.sa1
Reviewer #2 (Public review): https://doi.org/10.7554/eLife.94349.3.sa2
Reviewer #3 (Public review): https://doi.org/10.7554/eLife.94349.3.sa3
Author response https://doi.org/10.7554/eLife.94349.3.sa4

---

# Additional files

## Supplementary files
• MDAR checklist

## Data availability
All data generated or analyzed during this study are included in the manuscript and supporting files. Mean and median values and sample variability are listed in the manuscript as appropriate. Individual data points are represented on the plots/graphs. Source data have been deposited in Dryad (https://doi.org/10.5061/dryad.xksn02vqx).

The following dataset was generated:

| Author(s) | Year | Dataset title | Dataset URL | Database and Identifier |
|---|---|---|---|---|
| McLean et al. | 2024 | Cell-type-specific origins of locomotor rhythmicity at different speeds in larval zebrafish | https://doi.org/10.5061/dryad.xksn02vqx | Dryad Digital Repository, 10.5061/dryad.xksn02vqx |

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
