## [Editor Report · eLife assessment]

In this **fundamental** study, authors present **compelling** evidence for the diversity in cellular and synaptic properties of one class of spinal interneurons and tie it to their differentiated role in locomotor pattern generation. The findings reported here will be of broad interest to neuroscientists in general and to motor systems scientists in particular.

---

## [Referee Report · Reviewer #1 (Public review)]

Summary:

In this very interesting study, Agha and colleagues show that two types of Chx10-positive neurons (V2a neurons) have different anatomical and electrophysiological properties and receive distinct patterns of excitatory and inhibitory inputs as a function of speed during fictive swimming in the larval zebrafish. Using single cell fills they show that one cell type has a descending axon ("descending V2as"), while the other cell type has both a descending axon and an ascending axon ("bifurcating V2as"). In the Chx10:GFP line, descending V2as display strong GFP labeling, while bifurcating V2as display weak GFP labeling. The bifurcating V2as are located more laterally in the spinal cord. These two cell types have different electrophysiological properties as revealed by patch-clamp recordings. Positive current steps indicated that descending V2as comprise tonic spiking or bursting neurons. Bifurcating V2as comprise chattering or bursting neurons. The two types of V2a neurons display different recruitment patterns as a function of speed. Descending tonic and bifurcating chattering neurons are recruited at the beginning of the swimming bout, at fast speeds (swimming frequency above 30 Hz). Descending bursting neurons were preferentially recruited at the end of swimming bouts, at low speeds (swimming frequency below 30 Hz), while bifurcating bursting neurons were recruited for a broader swimming frequency range. The two types of V2a neurons receive distinct patterns of excitatory and inhibitory inputs during fictive locomotion. In descending V2as, when speed increases: (i) excitatory conductances increase in fast neurons and decreases in slow neurons; (ii) inhibitory conductances increase in fast neurons and increases in slow neurons. In bifurcating V2as, when speed increases: (i) excitatory conductances increase in fast neurons but does not change in slow neurons; (ii) inhibitory conductances increase in fast neurons and does not change in slow neurons. The timing of excitatory and inhibitory inputs was then studied. In descending V2as, fast neurons receive excitatory and inhibitory inputs that are in anti-phase with low contrast in amplitude and are both broadly distributed over the phase. The slow neurons receive two peaks of inhibition, one in anti-phase with the excitatory inputs and another just after the excitation. In bifurcating V2as, fast neurons receive two peaks of inhibition, while the slow ones receive anti-phase inhibition. They also show that silencing Dmrt3-labeled dI6 interneurons disrupted rhythm generation selectively at high speed.

Strengths:

This study focuses on the diversity of V2a neurons in zebrafish, an interesting cell population playing important roles in locomotor control and beyond, from fish to mammal. The authors provide compelling evidence that two subtypes of V2as show distinct anatomical, electrophysiological, speed-dependent spiking activity, and receive distinct synaptic inputs as a function of speed. This opens the door to future investigation of the inputs and outputs of these neurons. Finding ways to activate or inhibit specifically these cells would be very helpful in the years to come. The authors also provide an interesting speed-dependent circuit mechanism for rhythm generation.

Weaknesses:

No major weakness detected. The experiments were carefully done, and the data are of high quality.

---

## [Referee Report · Reviewer #2 (Public review)]

Summary:

Animals exhibit different speeds of locomotion. In vertebrates, this is thought to be implemented by different groups of spinal interneurons and motor neurons. A fundamental assumption in the field has been that neural mechanisms that generate and sustain the rhythm at different locomotor speeds are the same. In this study the authors challenge this view. Using rigorous in vivo electrophysiology during fictive locomotion combined with genetics, the authors provide a detailed analysis of cellular and synaptic properties of different subtypes of spinal V2a neurons that play a crucial role in rhythm generation. Importantly, they are able to show that speed related subsets of V2a neurons have distinct cellular and synaptic properties and maybe utilizing different mechanisms to implement different locomotor speeds.

Strengths:

The authors fully utilize the zebrafish model system and solid electrophysiological analyses to study active and passive properties of speed related V2a subsets. Identification of V2a subtype is based directly on their recruitment at different locomotor speeds and not on indirect markers like soma size, D-V position etc. Throughout the article, the authors have cleverly used standard electrophysiological tests and analysis to tease out different neuronal properties and link it to natural activity. For example, in Figures 2 and 4, the authors make comparisons of V2a spiking with current steps and during fictive swims showing spike rates measured with current steps are physiologically relevant and observed during natural recruitment. The experiments done are rigorous and well controlled.

The major claim of the manuscript is well substantiated by Figure 6 and 7. The authors have done rigorous experiments with statistical analysis to show that reciprocal inhibition is important for rhythmogenesis at fast speeds while recurrent inhibition is key at slow speeds. Furthermore, in Figure 7, a specific loss of reciprocal inhibition is shown to disrupt rhythmogenesis at high speeds but not at lower frequencies. These additions in the revised manuscript make the study extremely compelling.

The Discussion is well-written and does an excellent job in putting this current study in the context of what is previously known. The addition of a working model in Figure 8 does a great job in summing these exciting and novel findings.

Weaknesses:

None noted.

---

## [Referee Report · Reviewer #3 (Public review)]

The manuscript by Agha et al. explores mechanisms of rhythmicity in V2a neurons in larval zebrafish. Two subpopulations of V2a neurons are distinguishable by anatomy, connectivity, level of GFP, and speed-dependent recruitment properties consistent with V2a neurons involved in rhythm generation and pattern formation. The descending neurons proposed to be consistent with rhythm generating neurons are active during either slow or fast locomotion, and their firing frequencies during current steps are well matched with the swim frequency they firing during. The bifurcating (patterning neurons) are active during a broader swim frequency range unrelated to their firing during current steps. All of the V2a neurons receive strong inhibitory input but the phasing of this input is based on neuronal type and swim speed the neuron is active, with prominent in-phase inhibition in slow descending V2a neurons and bifurcating V2a neurons active during fast swimming. Antiphase inhibition is observed in all V2a neurons but it is the main source of rhythmic inhibition in fast descending V2a neurons and bifurcating neurons active during slow swimming. The authors suggest that properties supporting rhythmic bursting are not directly related to locomotor speed but rather to functional neuronal subtypes.

Strengths:

This is a well-written paper with many strengths including the rigorous approach. Many parameters, including projection pattern, intracellular properties, inhibition received, and activity during slow/fast swimming were obtained from the same neuron. This links up very well with prior data from the lab on cell position, birth order, morphology/projections, and control of MN recruitment to provide a comprehensive overview of the functioning of V2a interneuronal populations in the larval zebrafish. The added dI6 silencing experiments strengthen the claims made regarding the roles of reciprocal inhibition in rhythm and pattern at fast and slow speeds. The overall conclusions are well supported by the data.

Weaknesses:

The main weaknesses have been addressed in the revision.

---

## [Author Response]

The following is the authors’ response to the original reviews.

**eLife assessment**
The manuscript by Agha et al. provides a fundamental understanding regarding the participation of V2a interneurons in generating and patterning the locomotor rhythm. The authors provide convincing and solid evidence regarding the heterogeneity of V2a neurons in their intrinsic and synaptic properties and how these shape their outputs. The manuscript could be much improved by the inclusion of statistical analysis of some of the key data currently presented qualitatively.

We are extremely grateful for the positive and thorough comments provided by the three reviewers and have now had the opportunity to address all their concerns, as detailed below in our point-by-point response. Specifically, we have provided statistical analysis and major revisions to the text to help with rigor, clarity and interpretation, and we have also include new perturbation experiments that provide a more definitive test of one of our predictions – namely that reciprocal inhibition plays speed-specific roles in rhythm generation and pattern formation. The revisions greatly improve the manuscript and help bolster our conclusions.

Public Reviews:

**Reviewer #1 (Public Review):**
Summary:In this very interesting study, Agha and colleagues show that two types of Chx10-positive neurons (V2a neurons) have different anatomical and electrophysiological properties and receive distinct patterns of excitatory and inhibitory inputs as a function of speed during fictive swimming in the larval zebrafish. Using single-cell fills they show that one cell type has a descending axon ("descending V2as"), while the other cell type has both a descending axon and an ascending axon ("bifurcating V2as"). In the Chx10:GFP line, descending V2as display strong GFP labeling, while bifurcating V2as display weak GFP labeling. The bifurcating V2as are located more laterally in the spinal cord. These two cell types have different electrophysiological properties as revealed by patch-clamp recordings. Positive current steps indicated that descending V2as comprise tonic spiking or bursting neurons. Bifurcating V2as comprise chattering or bursting neurons. The two types of V2a neurons display different recruitment patterns as a function of speed. Descending tonic and bifurcating chattering neurons are recruited at the beginning of the swimming bout, at fast speeds (swimming frequency above 30 Hz). Descending bursting neurons were preferentially recruited at the end of swimming bouts, at low speeds (swimming frequency below 30 Hz), while bifurcating bursting neurons were recruited for a broader swimming frequency range. The two types of V2a neurons receive distinct patterns of excitatory and inhibitory inputs during fictive locomotion. In descending V2as, when speed increases: (i) excitatory conductances increase in fast neurons and decrease in slow neurons; (ii) inhibitory conductances increase in fast neurons and increase in slow neurons. In bifurcating V2as, when speed increases: (i) excitatory conductances increase in fast neurons but do not change in slow neurons; (ii) inhibitory conductances increase in fast neurons and do not change in slow neurons. The timing of excitatory and inhibitory inputs was then studied. In descending V2as, fast neurons receive excitatory and inhibitory inputs that are in anti-phase with low contrast in amplitude and are both broadly distributed over the phase. The slow neurons receive two peaks of inhibition, one in anti-phase with the excitatory inputs and another just after the excitation. In bifurcating V2as, fast neurons receive two peaks of inhibition, while slow ones receive anti-phase inhibition.Strengths:This study focuses on the diversity of V2a neurons in zebrafish, an interesting cell population playing important roles in locomotor control and beyond, from fish to mammals. The authors provide compelling evidence that two subtypes of V2as show distinct anatomical, electrophysiological, and speed-dependent spiking activity, and receive distinct synaptic inputs as a function of speed. This opens the door to future investigation of the inputs and outputs of these neurons. Finding ways to activate or inhibit specifically these cells would be very helpful in the years to come.Weaknesses:No major weakness was detected. The experiments were carefully done, and the data were of high quality.

We really appreciate the positive assessment and have addressed minor issues below.

**Reviewer #2 (Public Review):**
Summary:Animals exhibit different speeds of locomotion. In vertebrates, this is thought to be implemented by different groups of spinal interneurons and motor neurons. A fundamental assumption in the field has been that neural mechanisms that generate and sustain the rhythm at different locomotor speeds are the same. In this study, the authors challenge this view. Using rigorous in vivo electrophysiology during fictive locomotion combined with genetics, the authors provide a detailed analysis of cellular and synaptic properties of different subtypes of spinal V2a neurons that play a crucial role in rhythm generation. Importantly, they are able to show that speed-related subsets of V2a neurons have distinct cellular and synaptic properties and may utilize different mechanisms to implement different locomotor speeds.Strengths:The authors fully utilize the zebrafish model system and solid electrophysiological analyses to study the active and passive properties of speed-related V2a subsets. Identification of the V2a subtype is based directly on their recruitment at different locomotor speeds and not on indirect markers like soma size, D-V position etc. Throughout the article, the authors have cleverly used standard electrophysiological tests and analysis to tease out different neuronal properties and link it to natural activity. For example, in Figures 2 and 4, the authors make comparisons of V2a spiking with current steps and during fictive swims showing spike rates measured with current steps are physiologically relevant and observed during natural recruitment. The experiments done are rigorous and well-controlled.Weaknesses:The authors claim that a primary result of their study is that reciprocal inhibition is important for rhythmogenesis at fast speeds while recurrent inhibition is key at slow speeds. This is shown in Figure 6, however, the authors do not show any statistical tests for this claim. The authors also do not show any conclusive evidence that reciprocal inhibition is required for rhythmogenesis at fast speeds and vice versa for slow speeds. Additional experiments or modeling studies that conclusively show the necessity of these different inhibitory sources to the generation of different rhythms would be needed to strengthen this claim.

We have added new loss-of-function experiments as requested to strengthen the claim that reciprocal inhibition is critical for rhythmogenesis at fast speeds, but dispensable at slow. Specifically, we use botulinum toxin selectively expressed in Dmrt3-labeled dI6 interneurons, which play a role in reciprocal inhibition at a variety of speeds (new Figure 7). These experiments demonstrate a selective impact on rhythmic burst generation and alternation during periods of swimming where the highest frequency motor activity occurs. During lower frequency activity, rhythm generation is preserved, however motor output is selectively altered, consistent with the idea that reciprocal inhibition plays an important role in patterning at slow speeds.

The authors do a great job of teasing out cellular and synaptic properties in the different V2a subsets, however, it is not clear if or how these match the final output. For example, V2aD neurons are tonic or bursting for fast and slow speeds respectively but it is not intuitive how these cellular properties would influence phasic excitation and inhibition these neurons receive.

This question gets at the heart of what we are trying to illustrate in Figure 6. Specifically, in the new Figure 6E,F we have aligned the cumulative distribution of spikes recorded in cell-attached mode with phasic excitatory and inhibitory currents to reveal how well cellular properties versus patterns of synaptic drive match the final output (spikes). Our expectation was if intrinsic cellular properties where ultimately generating phasic spiking patterns, then patterns of excitatory and inhibitory drive need not be phasic. Instead, we see that synaptic drive is phasic with spiking occurring between peaks in excitation and troughs in inhibition. Since post-synaptic cellular properties should not impact the pre-synaptic excitation they receive, this suggests that phasic spiking in all V2a neurons regardless of the capacity for cellular rhythmogenesis is a result of phasic input. In response to this concern, we have elaborated our discussion of what cellular properties may contribute and the impact on output in the Discussion (L502-511).

It is not clear from the discussion why having different mechanisms of rhythm generation at different speeds could be an important circuit design. The authors use anguilliform and carangiform modes of swimming to denote fast and slow speeds but there are differences in these movements other than speed, like rostrocaudal coordination. The frequency and pattern of these movements are linked and warrant more discussion.

We appreciate the opportunity to elaborate on this point more in the Discussion. In particular, we have added more text to clarify differences in movement related to both pattern-formation and rhythm-generation (L373-398) and to also suggest potential reasons for differences in mechanisms of rhythm generation (L478-488).

**Reviewer #3 (Public Review):**
The manuscript by Agha et al. explores mechanisms of rhythmicity in V2a neurons in larval zebrafish. Two subpopulations of V2a neurons are distinguishable by anatomy, connectivity, level of GFP, and speed-dependent recruitment properties consistent with V2a neurons involved in rhythm generation and pattern formation. The descending neurons proposed to be consistent with rhythm-generating neurons are active during either slow or fast locomotion, and their firing frequencies during current steps are well matched with the swim frequency they firing during. The bifurcating (patterning neurons) are active during a broader swim frequency range unrelated to their firing during current steps. All of the V2a neurons receive strong inhibitory input but the phasing of this input is based on neuronal type and swim speed when the neuron is active, with prominent in-phase inhibition in slow descending V2a neurons and bifurcating V2a neurons active during fast swimming. Antiphase inhibition is observed in all V2a neurons but it is the main source of rhythmic inhibition in fast descending V2a neurons and bifurcating neurons active during slow swimming. The authors suggest that properties supporting rhythmic bursting are not directly related to locomotor speed but rather to functional neuronal subtypes.This is a well-written paper with many strengths including the rigorous approach. Many parameters, including projection pattern, intracellular properties, inhibition received, and activity during slow/fast swimming were obtained from the same neuron. This links up very well with prior data from the lab on cell position, birth order, morphology/projections, and control of MN recruitment to provide a comprehensive overview of the functioning of V2a interneuronal populations in the larval zebrafish. The overall conclusions are well supported by the data. Weaknesses are relatively minor and were largely related to terminology for some of the secondary conclusions.(1) The assumption is made that all in-phase inhibition is recurrent and out-of-phase inhibition is reciprocal. The latter is likely true but the definition of recurrent may be a bit loose as could be multisegmental feed-forward inhibition as well.

This is an excellent point, which was also raised by Reviewer 1. We have now added references that justify this assertion (L281-283). We also add a new figure with schematics (Figure 8) to make it clearer how we are defining sources of recurrent versus reciprocal inhibition, as based on the anatomical constraints of the circuit. We agree that multi-segmental inputs could contribute to inhibition, but they will likely be more broadly distributed based on rostro-caudal location and contribute to tonic sources of drive. We now clarify this (L285-286).

(2). In a few places, it is mentioned that the properties of the V2a-D neurons are consistent with pacemakers. This could be true of both the V2a-D and -B neurons that burst in response to depolarizing steps but the properties of the remaining (fast) V2a-D neurons do not seem to be consistent with pacemakers, based on the properties shown. Tonic firing at a frequency related to the locomotor speed the neuron is active during and strong antiphase inhibition may instead suggest a stronger network component driving the rhythmicity.

We have been purposefully agnostic regarding the relative contribution of pacemaking to rhythm generation in the paper. Our measurements of bursting overlap with swim frequencies only in the V2a-D subtype. Similarly, the spike rates of V2a-D neurons alone overlap with their swim frequencies (Fig 2D,G,I). Since both respond to tonic input (current injection) by spiking in a pattern that resembles their natural spiking behavior, we have treated these cellular properties both as pacemaking. Although the bursting behavior is more consistent with what is normally considered pacemaking in rhythmic motor circuits, in the basal ganglia field tonic firing of dopaminergic neurons in the substantia nigra is referred to as pacemaking. Since the tonic firing pattern overlaps with swimming frequency in the same way the bursting pattern does, we are less inclined to discount its possible contribution to rhythmogenesis based on the fact they do not burst. We have made modifications to the document to make this point clearer (L409-416). Regardless, our data argue that pacemaking is unlikely to be a major contributor to phasic firing in V2a neurons, at least at midbody, so we agree with you on this last point.

**Reviewer #1 (Recommendations For The Authors):**
I only have very minor suggestions.(1) It would be useful to add a table or a figure summarizing the main results (integration of anatomy, electrophysiological properties, synaptic inputs, firing, swimming speed).

We agree and have added a figure panel summarizing the main results (new Figure 8).

(2) Some statistics to possibly add (only suggestions): Do bifurcating V2as display significantly weaker GFP labeling than descending V2as? Do descending V2as have a significantly smaller soma size? Do descending V2as have a significantly lower rheobase and significantly higher resistance? Are tonic descending neurons and chattering bifurcating neurons located significantly more dorsally than the bursting descending and bifurcating neurons? Is there a way to show that bifurcating bursting neurons are recruited statistically on a broader swimming frequency range than other cell types (e.g. SD, coefficient of variation, cumulative distribution function with Kolmogorov-Smirnov test)?

For the first question, in all cases when we targeted more dimly labeled neurons they were bifurcating. We now clarify this in the text (L119, L129-132). However, this is difficult to quantify, since absolute levels of fluorescence will vary from preparation to preparation based on the dissection and intensity of epifluorescence illumination. In addition, we did not always take images prior to recording and levels of GFP after recording will vary depending on relative state of dialysis. So, unfortunately, we cannot provide a rigorous statistical analysis beyond the qualitative statement we provide.

For the remainder of the questions, we now provide statistical analysis for soma size, position, rheobase, and resistance for the data in Figure 2. Please note, we have reported all our statistical analyses in the figure legends. We also provide analysis of the density distributions of swimming frequencies for slow bursting bifurcating neurons and slow bursting descending neurons as requested, which are significantly different following a K-S test (L162).

(3) Some details to possibly add (only suggestions): proportion of neurons in which single cell fills were done/checked anatomically? Proportions of bursting/chattering/tonic/bursting neurons? In Figure 1, maybe define visually bifurcating vs descending neurons. In Figure 2I, the recruitment of bifurcating chattering neurons is not plotted. Is that normal? Figures 6D, E, maybe specify more clearly which neurons are the fast and slow ones. In Figure 3C, the X-axis name is missing.

For the first question, the proportion is 100%, since the morphology of all neurons was confirmed post recording, which we now clarify in the Methods section (L573). For the second question, the numbers of bursting/chattering/tonic/bursting neurons are now reported in legend of Figure 2, in addition to the total number of V2a-D and V2a-B types, so it is clear what proportion of the recording population this represents. For the third question, in Figure 1 we cannot define V2a neurons as bifurcating or descending yet, this was only possible to confirm after the recording (Figure 2), and was done for every neuron (as mentioned above). For the fourth question, for Figure 2I the chattering response was too variable to be meaningful in terms of averaging and plotting, which we now mention in the text (L169-171). The standard deviations are ridiculous. For the fifth question, we have modified Figures 6D, E to more clearly label fast and slow V2a neurons. Finally, we have included the X-axis label in Figure 3C, thank you!

(4) Some text to possibly modulate (only suggestions):A possible role for these V2a subtypes in the rhythm generation and pattern formation layer is an interesting idea but this may not be completely solved by the present experiments. Maybe the authors could suggest future experiments in the discussion that would establish how to tackle this important question (double bursts, deletions, etc...)?

We appreciate the opportunity to raise future experiments that could help further tease apart their contribution to rhythm and pattern and have now added potential experiments to the Discussion (L498-501; L527-529), which include more precise molecular identification, spatial perturbation, and computational modeling.

It would be nice to cite the references in which the rhythm/pattern CPG concept was proposed initially (lines 49-50 and elsewhere, Cf. Perret and Cabelguen 1980 Brain Res; Perret et al. 1989 Stance and Motion, Plenum Press; McCrea et al. 2006 J Physiol).

Apologies for our poor scholarship here, we now credit the appropriate primary research articles (L50-51).

In the abstract, it would be useful to say clearly which cells are descending vs. bifurcating ones. Same thing in the result section, maybe it would be nice to identify the two populations long before line 127.

We have modified the abstract and introduction sections accordingly. We also note that the two populations are defined in the first paragraph of the results (L90).

About the possible mechanism of rhythm generation, it is mentioned in line 54 that a single mechanism was proposed to exist, but the authors also mention in lines 122-123 that several mechanisms were proposed for rhythm generation... Maybe adjust the introduction?

As requested, we have clarified our meaning in the introduction (L55-58). Several mechanisms exist, but the likelihood that different mechanisms operate at different speeds has not been considered. Either cellular properties are tuned to different speeds (i.e., bursting is faster in neurons recruited at faster speeds) or network properties can explain different speeds (i.e., different frequencies and patterns emerge from the connectivity).

About the convention that in fish in-phase currents originate from the ipsilateral and out-of-phase currents originate from the contralateral side (lines 271-275), is there any reference for this assumption?

Yes, we now provide references (L281-283).

Lines 338-345 stating that reciprocal inhibition is important for rhythm generation as predicted by the half-center model can sound surprising to some authors considering that many studies showed that inhibition is not needed for rhythm generation, including lamprey hemicords stimulated electrically (Cangiano and Grillner 2003 J Neurophysiol; 2005 J Neurosci, Cangiano et al. 2012 Neuroscience), salamander hemicords or hemisegments stimulated chemically (Ryczko et al. 2010, 2015 J Neurophysiol), or rhythm activity evoked on each side of the cord using optogenetic stimulation of glutamatergic neurons (Hägglund et al. 2013 PNAS) etc. To demonstrate the importance of inhibition in rhythmogenesis, one would need to activate and/or deactivate the ipsilateral versus contralateral inhibitory neurons. It would be nice to maybe add citations to such studies if available in the zebrafish literature. Overall I would simply suggest modulating this section to be a bit more balanced conceptually.

We have included the above referenced studies for lampreys and added ones for tadpoles (L464-468), to stick with undulatory swimmers. We had focused on experiments with the most selective perturbations in the interests of space, but appreciate the opportunity to present both arguments. We also include new loss-of-function experiments that impact one spinal population linked to reciprocal inhibition (Dmrt3-labeled dI6 interneurons), which demonstrate a speed-specific impact on rhythmogenesis (L323-371; new Figure 7) and compare our findings to a recent study in the zebrafish literature examining the impact of spinal Dmrt3-ablations on axial rhythmogenesis (L426-433).

Line 676 "episodies".

Thanks, corrected.

**Reviewer #2 (Recommendations For The Authors):**
The authors make a claim that recurrent and reciprocal inhibition play key roles in rhythmogenesis at different speeds. This is not conclusively shown. Rayleigh's z-test can be used to test the significance of the directionality of circular data. Including more data from experiments or computational models to show the necessity of reciprocal or recurrent inhibition for timed spiking of V2a neurons would address this.

We have now modified Figure 6 so we can directly compare differences in reciprocal and recurrent inhibition between V2a types. We now report statistical analysis in the figure legends using a Watson’s Two Test for Homogeneity to test differences in the circular data. As mentioned above, we have also added new loss-of-function experiments as requested to strengthen the claim that reciprocal inhibition is critical for rhythmogenesis at fast speeds, but dispensable at slow. Specifically, we use botulinum toxin selectively expressed in Dmrt3-labeled dI6 interneurons, which play a role in reciprocal inhibition at a variety of speeds (new Figure 7). These experiments demonstrate a selective impact on rhythmic burst generation and alternation during periods of swimming where the highest frequency motor activity occurs. During lower frequency activity, rhythm generation is preserved, however motor output is selectively altered, consistent with the idea that reciprocal inhibition plays an important role in patterning at slow speeds.

In Figure 4D, the authors show that V2a neurons, both subtypes, spike in advance of the center of the motor burst. Recent studies (Jay et al., 2023) have shown differences in the timing of V2aD and V2aB neurons. Are there differences in the methods or selection of cells that would reflect differences in results?

This is a great point and we appreciate the opportunity to reconcile our observations here with those in Jay et al., 2023. In the Jay et al paper, we used drifting visual stimuli to evoke fictive swimming. These experiments allow you to uncouple rhythm generation (forward propulsion) and pattern formation (lateral direction). Notably, fictive swim frequencies during so called optomotor responses are below 35Hz, meaning that we are sampling exclusively from V2a neurons recruited during carangiform swim mode. In these experiments, slow V2a-D neurons fire well in advance of slow V2a-B neurons, compared to what we see here which is relatively synchronous. Critically, however, the phase-advanced firing pattern revealed in the Jay et al paper for V2a-D neurons aligns with the phase-advanced excitatory input reported here. In addition, the recruitment probabilities of slow V2a-D neurons are higher in the Jay et al paper than what we report here. Collectively these observations suggest either more effective excitation during optomotor responses (Jay et al) or more potent inhibition during escape responses (Agha et al). Ultimately, differences in the relative synchrony of firing among slow V2a-D and slow V2a-B neurons appears to depend on the nature of the stimulus and range of swim frequencies, where in one case frequency and amplitude modulation are coupled over a broad range of frequencies (somatosensory stimuli delivered here), while in the other case frequency and amplitude modulation are uncoupled over a narrow range of frequencies (visual stimuli in Jay at al). We now elaborate on this point in the Discussion (L485-498).

Given the conserved nature of spinal circuits across vertebrates, it is also important to discuss these findings in the context of limbed animals. In tetrapods, changes in locomotor speed also involve pattern/gait changes, however, it is not known if or how these changes in frequency and pattern are linked. This study, by suggesting that different speeds are implemented not only by different neurons but possibly by different neuronal mechanisms, provides important cues for the missing link and would strengthen the discussion.

We agree and have made substantial edits to the beginning Discussion to provide better context for the impact of our work (L373-398).

Minor points:Line 122: of needs to be replaced by or.

Corrected, thanks!

Figure 3B Top panel: What is the grey bar?

This has been removed for clarity.

Figure 3B bottom panel is not referenced in the main text at all.

Now referenced (L187, L189)

Line 260: 2nd inhibition needs to be replaced with excitation.

Done, thanks!

**Reviewer #3 (Recommendations For The Authors):**
Minor comments:- Figure 2 panel ordering is visually appealing but tough to follow.

We apologize and tried reconfigurations, but they just looked too kludgy. Hoping for a pass on this one.

- Lines 164-166 and 319-327 (related to comment 2 above): For the fast/tonic V2a-Ds, it is not clear that this is intrinsic and it is not consistent with pacemaker properties. This could also be (and likely is) synaptically/network-driven rhythmicity, although the firing frequencies match up well with the swim frequencies.

Fast/tonic V2a-Ds were tested with somatic current injection as with all other neurons, which we assume primarily reflects intrinsic cellular properties. The spike rates we observe in fast/tonic V2a-Ds overlap with spike rates observed during fictive swimming, so they are positioned as well as bursting neurons to contribute to pacemaking. We also elaborate on this point in response to Major Comment #2.

- Lines 189-192: The patterning neurons receive excitatory drive before rhythm-generating neurons. The time constant explanation makes sense for why two neurons with a common drive would fire at different times but this does not support the proposed hierarchical arrangement or being consistent with V2a-Bs being downstream as mentioned in lines 49-56 and 218-219.

In response to this point, we have modified Figure 6 so we can directly compare the timing of presynaptic excitatory inputs between the types. Here it can be seen clearly that phasic excitatory inputs to both fast and slow V2a-Ds are phase-advanced relative to fast and slow V2a-Ds (Figure 6B,C). As the reviewer mentions, it is likely a combination of time constants and the relative balance of excitation and inhibition that ultimately lead to synchronous spiking despite differences in the timing of inputs.

- Lines 338-339: It is not shown that the rhythm relies on inhibition during slow.

This line has been removed in the revision process.

- Consistent with the importance of reciprocal (contralateral) inhibition in fast locomotion here, rodent fictive locomotion is slower in hemisect than in the full cord. However, the Rybak and O'Donovan groups suggest that this is due to loss of drive to ipsilateral inhibitory neurons by excitatory contralateral projections, rather than contralateral inhibitory interneurons (see Falgairolle and O'Donovan 2019, 2021, and Shevtsova et al 2022).

This is an interesting point that highlights how we are defining reciprocal versus recurrent inhibition. In this example, although ipsilaterally-projecting interneurons are responsible for inhibition, since they are excited by commissurally-projecting excitatory interneurons, we would classify this as feedforward (reciprocal) not feedback (recurrent) inhibition. So reciprocal (feedforward) inhibition is still important to get higher frequency rhythms, it is di-synaptic in this case. We have added a new figure (Figure 8) to clarify what we mean by reciprocal (feedforward) and recurrent (feedback) based on the ipsilateral projection patterns of V2a neurons, and point out the definitions would be flipped for excitatory interneurons in the Discussion (L452-455).